# "What Data Benefits My Classifier?" Enhancing Model Performance and Interpretability through Influence-Based Data Selection

**Anshuman Chhabra**[†]**, Peizhao Li**[§‡]**, Prasant Mohapatra**[†]**, and Hongfu Liu**[§]

[†] University of California, Davis      [§] Brandeis University     [‡] GE HealthCare

{chhabra,pmohapatra}@ucdavis.edu    {peizhaoli,hongfuliu}@brandeis.edu

## Abstract

Classification models are ubiquitously deployed in society and necessitate high utility, fairness, and robustness performance. Current research efforts mainly focus on improving model architectures and learning algorithms on fixed datasets to achieve this goal. In contrast, in this paper, we address an orthogonal yet crucial problem: given a fixed convex learning model (or a convex surrogate for a non-convex model) and a function of interest, we assess what data benefits the model by interpreting the feature space, and then aim to improve performance as measured by this function. To this end, we propose the use of influence estimation models for interpreting the classifier's performance from the perspective of the data feature space. Additionally, we propose data selection approaches based on influence that enhance model utility, fairness, and robustness. Through extensive experiments on synthetic and real-world datasets, we validate and demonstrate the effectiveness of our approaches not only for conventional classification scenarios, but also under more challenging scenarios such as distribution shifts, fairness poisoning attacks, utility evasion attacks, online learning, and active learning.

## 1 Introduction

Machine learning models have become essential tools in various societal applications for automating processes and generating insights (Pouyanfar et al., 2018; Liu et al., 2017). Along with the choice of model type/architecture, data is a critical component of the learning process, and the quality and quantity of training data have significant impacts on the model performance (Blum & Langley, 1997). Despite this, current research mainly focuses on proposing high-performing model architectures or learning approaches while keeping the training data fixed. However, it is evident that not every sample in the training set augments model performance. Furthermore, the same data sample could be beneficial or detrimental to performance depending on the type of model being used (Blum & Langley, 1997). Therefore, in this paper, we aim to answer the research question "*what data benefits the learning model in a certain aspect?*" and select suitable training data to improve model performance.[1]

Our work relates to but contrasts with research on *data valuation*. Data valuation aims to assign a monetary value or worth to a particular set or collection of data, whereas our goal in this work is to analyze what type of data can be utilized to enhance a model. Data valuation can be performed in a variety of ways, such as using cooperative game theory (Shapley-value or Banzhaf index) (Ghorbani & Zou, 2019; Jia et al., 2019; Kwon & Zou, 2022; Ghorbani et al., 2020; Wang & Jia, 2023) and reinforcement learning (Yoon et al., 2020). It is also important to note that as alternative to influence functions, Shapley-value based data valuation approaches (Ghorbani & Zou, 2019; Jia et al., 2019) can also be used in our algorithms. However, Shapley-value based approaches such as TMC-Shapley (Ghorbani & Zou, 2019) require the model to be retrained and evaluated multiple times, where even the most efficient known non-distributional model-specific algorithms (Jia et al., 2019) require $\mathcal{O}(\sqrt{n}\log(n)^2)$ model retraining steps, and $n$ is the number of training samples. Other ap-

---

[1] With respect to utility/accuracy, fairness, and adversarial robustness.

proaches such as KNN-Shapley (Jia et al., 2018) that are computationally efficient are model-agnostic and designed to work with utility. Hence, KNN-Shapley is neither tailored to the specific model being used nor a function of interest other than utility (such as fairness and robustness), and cannot guarantee performance on the downstream classifier (Blum & Langley, 1997). Moreover, current work on Shapley-value based frameworks generally estimate the data value in utility, with either very few or no work studying fairness (Arnaiz-Rodriguez et al., 2023) and robustness. Reinforcement learning-based data valuation (Yoon et al., 2020) faces similar issues due to its complex design and poor convergence.

Our work is also closely related to other work on *data influence*, such as the seminal paper on influence functions by Koh and Liang (Koh & Liang, 2017), follow-up work such as TracIn (Pruthi et al., 2020), *representer point* (Yeh et al., 2018), and Hydra (Chen et al., 2021). However, these methods are fundamentally influence estimation approaches, and do not directly answer our research question, which is to identify and interpret the feature space for improving the model's performance. Other miscellaneous approaches, such as *Datamodels* (Ilyas et al., 2022) only work with one specific target sample at a time, posing a limitation. Unlike all these methods, our data selection approaches can also uniquely handle scenarios where data is unlabeled such as in active learning applications. Moreover, our work considers utility, fairness, and robustness under diverse application scenarios. Note that our data selection strategy can even boost the performance of methods such as the influence-based data reweighing approach of (Li & Liu, 2022) for improving fairness (Section 5.1) and outperform existing influence-based approaches for active learning (Liu et al., 2021) (Section 5.5). Thus, we utilize influence functions as a tool (allowing for any influence approach to be used interchangeably) and extend its applicability to diverse domains and scenarios not considered previously.

Finally, our work also conceptually relates to several other data-related areas. *Data efficiency* approaches (Paul et al., 2021; Coleman et al., 2020; Mirzasoleiman et al., 2020; Shen & Sanghavi, 2019; Killamsetty et al., 2021; Jain et al., 2022) aim to accelerate deep model training by pruning or selecting a subset of data, which is beyond the scope of our work. *Feature selection* approaches (Cai et al., 2018; Hall, 1999) aim to select important features for training, but are limited in scope as they only work for utility. *Active learning* (Cohn et al., 1996; Wei et al., 2015) partially aligns with our research question, as it involves selecting unlabeled data points to be annotated and added to the training set for improving model performance. However, its applicability is limited to this specific scenario, while our work considers a broader range of applications, including this one. *Antidote data* methods (Rastegarpanah et al., 2019; Chhabra et al., 2022b; Li et al., 2023) aim to add generated data to the training set for mitigating unfairness but cannot be used to interpret the usefulness of existing samples.

**Contributions**. We summarize our major contributions as follows:
- We utilize influence functions to assess what data improves a given convex classifier (or a surrogate for a non-convex model) with respect to utility, fairness, and robustness by interpreting the feature space. Our key idea is to use tree-based influence estimation models to understand and interpret which sample features contribute positively or negatively to the model's performance with respect to desired evaluation functions on a validation set. Additionally, we design a data selection strategy to achieve performance improvements.
- We verify the correctness of our proposed approaches to improve the classification model's utility, fairness, and robustness on synthetic data and demonstrate the usefulness of our approaches on real-world diverse datasets with regard to efficacy, simplicity, and interpretability.
- We move beyond the above classical classification setting and apply our approaches to diverse and practical application scenarios, including correcting for fairness distribution shift, combating fairness poisoning attacks, defending against adaptive evasion attacks, online learning with streaming batch data, and analyzing unlabeled sample effectiveness in active learning.

## 2 PRELIMINARIES AND BACKGROUND

We first introduce influence functions on convex models (logistic regression) and how to measure sample influence on utility, fairness, and robustness. We then discuss extensions to non-convex models.

**Influence Functions.** Given a training set $Z = \{(x_i, y_i)\}_{i=1}^n$ and a classifier trained using empirical risk minimization by a loss function $\ell$, we have the optimal parameters of the classifier $\hat{\theta} = \arg\min_{\theta \in \Theta} \frac{1}{n} \sum_{i=1}^n \ell(x_i, y_i; \theta)$. Influence functions (Koh & Liang, 2017) measure the effect of changing an infinitesimal weight of samples on a validation set, based on an impact function $f$ evaluating the quantity of interest, such as utility, fairness, and robust-

ness. Downweighting a training sample $x_j$ by a very small fraction $\epsilon$ leads the model parameter to $\hat{\theta}(x_j; -\epsilon) = \arg\min_{\theta \in \Theta} \frac{1}{n} \sum_{i=1}^{n} \ell(x_i, y_i; \theta) - \epsilon\ell(x_j, y_j; \theta)$. The actual impact of such a change can be written as $\mathcal{I}^*(x_j; -\epsilon) = f(\hat{\theta}(x_j; -\epsilon)) - f(\hat{\theta})$, where $f(\hat{\theta}(x_j; -\epsilon))$ can be obtained by retraining the model without $x_j$. Assuming $l$ is strictly convex and twice differentiable, and $f$ is also differentiable, the actual impact can be estimated without expensive model retraining by $\mathcal{I}(x_j; -\epsilon) = \lim_{\epsilon \to 1} f(\hat{\theta}(x_j; -\epsilon)) - f(\hat{\theta}) = \nabla_{\hat{\theta}} f(\hat{\theta})^\top \mathbf{H}_{\hat{\theta}}^{-1} \nabla_{\hat{\theta}} \ell(x_j, y_j; \hat{\theta})$, where $\mathbf{H}_{\hat{\theta}} = \sum_{i=1}^{n} \nabla_{\hat{\theta}}^2 \ell(x_j, y_j; \hat{\theta})$ is the Hessian matrix of $\ell$ and is invertible since $\ell$ is assumed to be convex. Thus, we can measure the influence of removing sample $x_j$ from the training set as $\mathcal{I}(-x_j)$.

**Measuring Influence on Utility.** If we instantiate the impact function $f$ to calculate the loss value on a validation set $V$, we can measure the training sample influence on utility as follows:

$$\mathcal{I}^{\text{util}}(-x_i) = \sum_{(x,y) \in V} \nabla_{\hat{\theta}} \ell(x, y; \hat{\theta})^\top \mathbf{H}_{\hat{\theta}}^{-1} \nabla_{\hat{\theta}} \ell(x_i, y_i; \hat{\theta}). \tag{1}$$

**Measuring Influence on Fairness.** Similarly, we can instantiate the impact function $f$ by group fairness (Dwork et al., 2012), such as demographic parity (DP) or equal opportunity (EOP) to measure influence on fairness. Consider a binary sensitive attribute defined as $g \in \{0, 1\}$ and let $\hat{y}$ denote the predicted class probabilities. The fairness metric DP is defined as: $f^{\text{DP-fair}}(\hat{\theta}, V) = \left| \mathbb{E}_V[\hat{y}|g = 1] - \mathbb{E}_V[\hat{y}|g = 0] \right|$. Based on that, we can calculate the training sample influence on fairness as follows:[2]

$$\mathcal{I}^{\text{DP-fair}}(-x_i) = \nabla_{\hat{\theta}} f^{\text{DP-fair}}(\hat{\theta}, V)^\top \mathbf{H}_{\hat{\theta}}^{-1} \nabla_{\hat{\theta}} \ell(x_i, y_i; \hat{\theta}). \tag{2}$$

**Measuring Influence on Adversarial Robustness.** We can also measure which points contribute to adversarial robustness (or vulnerability) using influence functions. To do so, we first define an *adversary*– any attack approaches to craft adversarial samples that can be used including black-box, white-box, among others. We consider a white-box adversary (Megyeri et al., 2019) specific to linear models, which can be easily extended to other models and settings, such as FGSM (Goodfellow et al., 2014), PGD (Madry et al., 2017), etc. To craft an adversarial sample, we take each sample $x$ of the validation set $V$ and perturb it as $x' = x - \gamma \frac{\hat{\theta}^\top x + b}{\hat{\theta}^\top \hat{\theta}} \hat{\theta}$, where $\hat{\theta} \in \mathbb{R}^d$ are the linear model coefficients, $b \in \mathbb{R}$ is the intercept, and $\gamma > 1$ controls the amount of perturbation added. Since the decision boundary is a hyperplane, we simply move each sample orthogonal to it by adding minimal perturbation. In this manner, we can obtain an adversarial validation set $V'$ which consists of $x'$ for each sample $x$ of $V$. The class labels $y$ remain unchanged. Now, we can compute adversarial robustness influence for each training sample as follows:

$$\mathcal{I}^{\text{robust}}(-x_i) = \sum_{(x',y) \in V'} \nabla_{\hat{\theta}} \ell(x', y; \hat{\theta})^\top \mathbf{H}_{\hat{\theta}}^{-1} \nabla_{\hat{\theta}} \ell(x_i, y_i; \hat{\theta}). \tag{3}$$

**Extension to Non-Convex Models**. A current limitation of influence functions is that they require the model to satisfy strict convexity conditions, implying its Hessian is positive definite and invertible, and that it is trained to convergence (Koh & Liang, 2017). To extend influence functions to non-convex models, several possible solutions exist: (1) a linear model can be used as a surrogate on the embeddings obtained via the non-convex model (Li & Liu, 2022); (2) a damping term can be added to the non-convex model such that its Hessian becomes positive definite and invertible (Han et al., 2020); and (3) for certain tasks specific or second-order influence functions can be derived (Basu et al., 2020b; Alaa & Van Der Schaar, 2020). In this paper, we adopt the above first strategy for non-convex models.

Note that there is work that shows that influence functions for deep learning models are *fragile* (Basu et al., 2020a). However, follow-up work (Bae et al., 2022; Epifano et al., 2023) has contrasted with some of these findings, resulting in the applicability of influence functions in deep learning being a contentious research topic. Despite this, multiple work has demonstrated their benefits in deep networks, such as on BERT (Han et al., 2020), ResNets (Liu et al., 2021; Yang et al., 2022), CNNs (Koh & Liang, 2017; Schioppa et al., 2022). We also provide preliminary results using our data selection approach on BERT to show this (Appendix H). However, our paper is primarily focused on convex models and resolving the undecided research question of deep learning influence fragility is not our goal. Thus, we resort to using a surrogate convex model on the embedding space of the non-convex model. This strategy subsequently obtains positive experimental performance as well.

---

[2]We provide the sample influence definitions for EOP in Appendix A.

## 3 PROPOSED APPROACHES

We now present Algorithm 1 for influence estimation via trees to interpret how data samples and feature ranges impact model performance with respect to utility, fairness, and robustness. Additionally, we propose Algorithm 2 for trimming training samples to improve model performance given a budget.

### 3.1 ESTIMATING THE INFLUENCE OF SAMPLES

Influence functions can efficiently estimate the data impact in various aspects. To further provide their interpretations, we employ decision trees to uncover which sample features contribute positively or negatively to the model's performance with respect to desired evaluation functions. To address the issue of the tree depth on interpretability, we utilize hierarchical shrinkage (Agarwal et al., 2022) to regularize the tree. We present our approach for influence estimation in Algorithm 1, which takes as input a regularization parameter $\lambda$, training set $Z$, validation set $V$, and an influence function definition $\mathcal{I}^{\mathcal{F}}$ for samples in $Z$ on $V$. Specifically, we iterate over each training set tuple $(x_i, y_i) \in Z$ and create a new regression dataset $M$ where the regressor is $[x_i, y_i]$ (block matrix notation implies appending $y_i$ to the end of $x_i$) and the response variable is computed via $\mathcal{I}^M(-x_i)$. Note that we append the label to our dataset since influence estimation is dependent on class labels.

Then, we train a regression tree $h$ using CART (Breiman et al., 2017). To ensure that the tree is interpretable while preserving performance, we utilize hierarchical shrinkage post-training. For our tree $h$ and for a given sample $q_i$ in the dataset $M$, let its leaf-to-root node path in the tree denote as $t_w \subset t_{w-1} \subset ... \subset t_0$. Here $t_w$ represents the leaf node and $t_0$ is the root node. Then we define two mapping functions for ease of readability: $\phi$ and $\xi$. The function $\phi$ takes as input a tree node and returns the number of samples it contains.

---

**Algorithm 1** : Influence Estimation Via Trees

**Input**: Training set $Z$, Validation set $V$, Influence function $\mathcal{I}^{\mathcal{F}}$, Hyperparameter $\lambda$
**Output**: Influence Estimator Tree $\hat{h}$
1: **initialize** $M \leftarrow \emptyset$.
2: **for** $(x_i, y_i) \in Z$ **do**
3:      $q_i \leftarrow [x_i, y_i]$ //appending label to $x_i$
4:      $M \leftarrow M \cup \{(q_i, \mathcal{I}^{\mathcal{F}}(-x_i))\}$
5: **end for**
6: **train** $h$ using CART (Breiman et al., 2017) on $M$.
7: **return** $\hat{h}$ by using hierarchical shrinkage (Agarwal et al., 2022) on $h$ with $\lambda$.

---

The function $\xi$ takes as input the query sample $q$ and the tree node $t$ and outputs the average predictive response for $q$ at node $t$. The overall regression tree prediction model for $q_i$ can then be written as: $h(q_i) = \xi(q_i, t_0) + \sum_{j=1}^{w} \xi(q_i, t_j) - \xi(q_i, t_{j-1})$. Hierarchical shrinkage regularizes the tree $h$ by shrinking the prediction over each tree node by the sample means of its parent nodes, $\hat{h}(q_i) = \xi(q_i, t_0) + \sum_{j=1}^{w} \frac{\xi(q_i, t_j) - \xi(q_i, t_{j-1})}{1 + \lambda/\phi(t_{j-1})}$.

### 3.2 DATA TRIMMING FOR SUPERVISED MODELS

We present Algorithm 2 for trimming training datasets which takes input as before $Z, V, \mathcal{I}^{\mathcal{F}}$ for samples in $Z$ on $V$, and a budget $b$ for the number of samples to remove and outputs the trimmed dataset $Z'$. The goal is to remove samples from the dataset that have negative influence. First, we initialize the sets $J, K$, and $Z'$. Then we populate $J$ with the influence values of samples in $J$, and $K$ with the indices of these samples (lines 2-3). We then sort $J$ in order of increasing positive influence, and $K$ according to the sorted order obtained via $J$. In $K$ we only care about the first $\min\{b, b'\}$ indices (line 9)

---

**Algorithm 2** : Data Trimming

**Input**: Training set $Z$, Validation set $V$, Influence function $\mathcal{I}^{\mathcal{F}}$, Budget $b$
**Output**: Trimmed Dataset $Z'$
1: **initialize** $J \leftarrow \emptyset, K \leftarrow \emptyset, Z' \leftarrow \emptyset$.
2: **for** $(x_i, y_i) \in Z$ **do**
3:      $J \leftarrow J \cup \{\mathcal{I}^{\mathcal{F}}(-x_i)\}$ //on set $V$
4:      $K \leftarrow K \cup \{i\}$
5: **end for**
6: **sort** $J$ in ascending order.
7: **sort** $K$ using $J$.
8: $b' \leftarrow \sum \mathbb{1}_{j<0, j \in J}$.
9: $Z' \leftarrow Z' \cup \{x_i\}, \forall i \notin K_{:\min\{b, b'\}}, x_i \in Z$.
10: **return** $Z'$.

---

where $b'$ is the total number of negative influence samples (line 8). Finally, we select only those samples to be part of $Z'$ that do not have indices in this subset of $K$ and return $Z'$.

## 4 EXPERIMENTAL RESULTS

In this section, we present results for our algorithms presented in the previous section. We first verify the correctness of our algorithms on synthetically generated toy data. We analyze how our influence

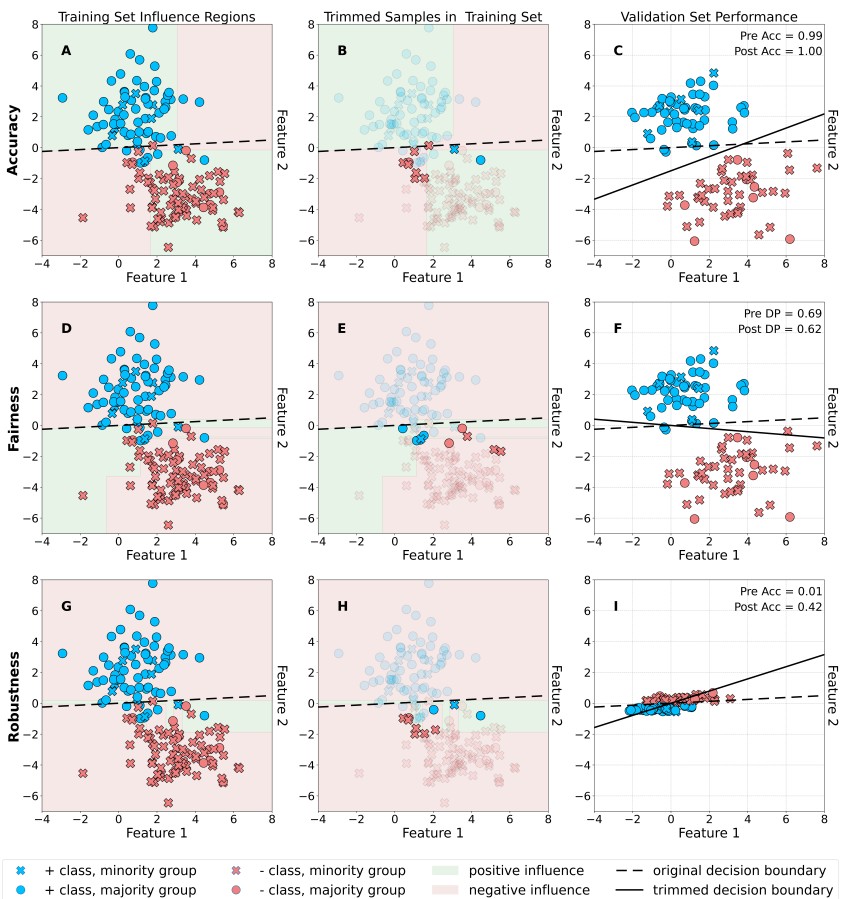

Figure 1: Correctness verification on toy data for utility (accuracy), fairness, and robustness functions. (+) class samples are denoted in blue, (-) class samples are denoted in orange, majority group samples are denoted by ○ and minority group samples are denoted by ×. A, D, and G show the training set with the original logistic regression decision boundary, and we visualize the influence regions generated by our tree model from Algorithm 1 as positive (light green) or negative (light red) for each function: utility/fairness/robustness. B, E, and H denote the points to be trimmed as identified by Algorithm 2 for each function type. C and F denote the validation set and how we obtain improved accuracy and fairness performance on it after trimming. I denotes the adversarial validation set and how post trimming we can improve performance on adversarial data.

estimation algorithm can be used to visualize and interpret regions of positive or negative influence, and trim the training set to improve accuracy, fairness, and robustness on the validation set. We then demonstrate the effectiveness of our algorithms on test sets of four real-world datasets and greatly improve their performance, especially for fairness and robustness.

**Correctness Verification on Toy Data.** We generate 150 train and 100 validation samples using two isotropic 2D-Gaussian distributions and use logistic regression for binary classification. We then analyze the training sample's positive/negative impact on model's accuracy, fairness and robustness in Figure 1 A, D, and G, respectively. The green regions denote the positive influences, while the pink regions denote the negative influences. These regions show the feature values derived from training samples that affect the linear model either positively or negatively for the function of interest. In Figure 1 A, most samples contribute positively for utility, and hence lead to positive influence regions. Similarly, in Figure 1 D, most of the samples are harmful for fairness as the (+) class has a large number of majority group samples and the (-) class has a large number of minority samples. Thus, most of the regions identified are of negative influence on fairness. Subsequently, we use Algorithm 2 with a budget $b = 10$ to obtain the points to be trimmed, shown in Figure 1 B, E, and H. We then remove these points from training set, resulting in significant performance improvement on the validation set for each function of interest, as shown in Figure 1 C, F, and I. Notably, the robustness is significantly improved from

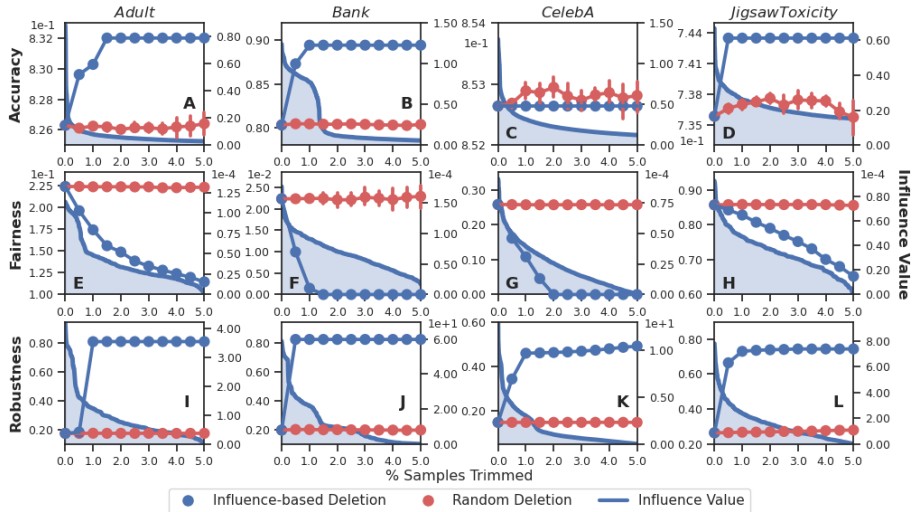

Figure 2: Double y-plot of our trimming method on the test sets of *Adult*, *Bank*, *CelebA*, and *Jigsaw Toxicity*. The left y-axes present algorithmic accuracy, fairness, and robustness, while the right y-axes present the influence value of the trimmed samples. Higher values indicate better accuracy and robustness, while lower values indicate better fairness. The blue lines represent the performance of our trimming method, while the red lines represent the performance with random trimming.

0.01 to 0.42 by simply removing the identified points. Additionally, Figure 1 I shows an adversarial set with samples perturbed in Section 2, where adversarial points crowd around the original decision boundary and are difficult to classify accurately. Note that while validation of influence computations has been well-demonstrated in previous studies (Koh & Liang, 2017; Li & Liu, 2022), we provide comparisons for actual sample influence via model retraining and estimated influence in Appendix B.

**Algorithmic Performance on Real-World Datasets.** We demonstrate the performance of our proposed methods on four real-world datasets, consisting of two tabular datasets *Adult* (Kohavi et al., 1996) and *Bank* (Moro et al., 2014), one visual dataset *CelebA* (Liu et al., 2018), and one textual dataset *Jigsaw Toxicity* (Noever, 2018). Details on datasets and training are present in Appendix C. Influence is measured on the validation set and logistic regression is the base model used. We present results on test sets of these datasets in Figure 2, along with influence value magnitudes corresponding to trimmed samples for each function type. Results with using MLP as the base model are shown in Appendix D.1 and exhibit similar trends. While full visualizations of the tree estimators (Algorithm 1) for all four datasets are presented in Appendix F, as a descriptive example we provide the left subtree for fairness influence on *Jigsaw Toxicity* in Figure 3.

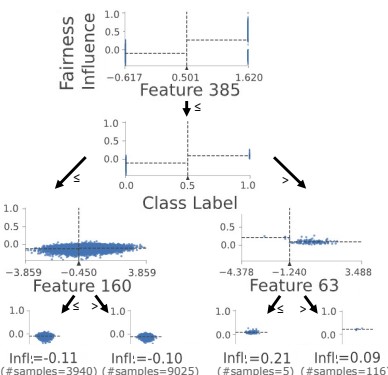

Figure 3: Example influence subtree.

In Figure 2, we demonstrate the effect of increasing the budget $b$ of our trimming approach (up to 5% of training data) and compare it to a random baseline that removes $b$ training samples.[3] The results show that our methods are able to improve fairness (DP)[4] and robustness significantly, while the random baseline fails to do so. This is particularly evident for datasets where large fairness disparities exist. Additionally, we are able to achieve significant improvements in accuracy on the adversarial test set for all datasets, indicating that our trimming can improve the model's robustness by effectively selecting training samples. However, we observe that trimming does not lead to much improvement in accuracy for most datasets, except for *Bank*, where we obtain a 10% increase. We conjecture that

---

[3]We fail to wholly compare with model-specific Shapley-value data valuation methods such as TMC-Shapley (Ghorbani & Zou, 2019) in utility due to their prohibitively long execution time on our datasets. However, we provide some preliminary results on subsampled data and execution time comparisons in Appendix E.

[4]Additional experiments using EOP as the fairness metric are presented in Appendix D.2.

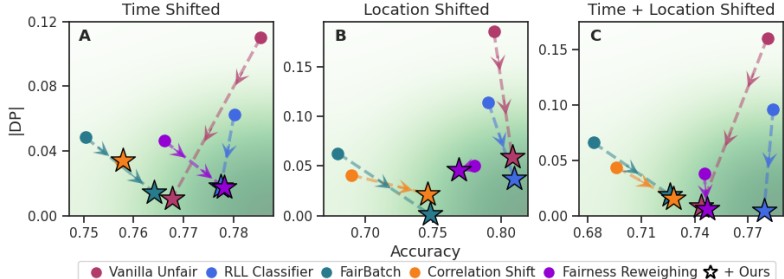

Figure 4: Performance of fairness and utility of several fairness algorithms under distribution shift. A greener background denotes a better solution in terms of both accuracy and fairness.

the current logistic regression models may already achieve optimal performance on these datasets, as seen by the quick decrease in utility influence value, which suggests little room for improvement. This also implies that we may not require as much training data to achieve similar utility results.

# 5 APPLICATION SCENARIOS BEYOND CONVENTIONAL CLASSIFICATION

We extend the conventional classification setting of the previous section by conducting extensive experiments for various practical deployment and maintenance scenarios, such as correcting distribution shift unfairness, combating (fairness) poisoning and (utility) evasion attacks, online learning with streaming batch data, and analyzing unlabeled sample effectiveness for active learning. We thus show the effectiveness of our proposed methods in improving model performance across diverse application scenarios either by trimming the training set, or by identifying new beneficial samples to add to it.

## 5.1 MITIGATING UNFAIRNESS UNDER DISTRIBUTION SHIFT

We demonstrate how our proposed method can serve as an effective fairness intervention and enhance the current methods under distribution shift. Fairness-specific distribution shifts (Roh et al., 2023) occur when the distribution of sensitive attributes (and class labels) between the training and test sets have drifted apart, resulting in severe issues of unfairness on the test set. Although it is a newly emerging direction, some pioneering attempts (Roh et al., 2023; Maity et al., 2021; Giguere et al., 2022) have been taken to address the change in bias between the original and test distribution. Typically, these methods have access to some samples from the test set to study the change in bias. In our experiments, these samples can constitute the validation set $V$ given that the validation and test set are correlated with each other. Here we employ multiple fair intervention approaches as baselines– such as Correlation Shift (Roh et al., 2023), FairBatch (Roh et al., 2021), Robust Fair Logloss Classifier (RLL) (Rezaei et al., 2020), Influence-based Reweighing (Li & Liu, 2022), and vanilla logistic regression. We use the *ACS Income* (Ding et al., 2021) dataset and construct three different distribution shift scenarios– time-based, location-based, time&location-based. In all three scenarios, we use the same training data constituting California (2014) data with 18000 samples, while the test sets consist of 6000 different samples: for time-based shift, this comprises California (2018) data, for location-based shift, it is Michigan (2014) data, and for the time&location-based shift, we use Michigan (2018) data. We set budget $b = 600$ which constitutes only 3.33% of the training set.

We present results as fairness-accuracy plots in Figure 4. We show that our trimming based approach can greatly improve fairness under distribution shift even when utilized with the vanilla model, making it competitive with the fairness intervention baselines. Moreover, except for the vanilla model (Figure 4 A&C), our trimming approach with the fairness intervention methods improves their accuracy significantly as well. This demonstrates the generalizability of our approach, as it can be used as a pre-processing step to boost the performance of other mitigation strategies.

## 5.2 COMBATING POISONING ATTACKS AGAINST FAIRNESS

We use our trimming approach as an effective defense for mitigating poisoning attacks against fairness. Recently, several attacks (Solans et al., 2021; Mehrabi et al., 2021; Chhabra et al., 2023a) have been proposed to reduce fairness of learning models. Random Anchoring Attack (RAA) and Non-random Anchoring Attack (NRAA)(Mehrabi et al., 2021) are two representative ones. These first select a set of target points to attack, and then poison a small subset of the training data by placing additional

anchoring points near the targets with the opposite class label. The target samples are uniformly randomly chosen in RAA, while NRAA selects "popular" target samples that result in the maximization of bias for all samples in the test set post the attack. Here we follow the original implementations of RAA and NRAA and use their same datasets including *German* (Hsieh, 2005), *Compas* (Brennan et al., 2009), and *Drug* (Fehrman et al., 2017). Since all these are different sizes we set the trimming budget $b \leq 10\%$ of training samples and the attacker's poisoning budget is $10\%$ of training samples.

Table 1 shows the results obtained. Our trimming approach is useful for combating fairness-reducing poisoning attacks, as it improves fairness (and utility for *Compas*) performance compared to the metric values obtained before and after the attacks. It is also significant to note that there are currently no well-performing defenses proposed for the aforementioned attack in the supervised setting (Chhabra et al., 2023a). Our trimming approach thus shows promise as

Table 1: Performance of fairness poisoning attacks

| Method | German | | Compas | | Drug | |
|---|---|---|---|---|---|---|
| | DP | ACC | DP | ACC | DP | ACC |
| Before | 0.117 | **0.670** | -0.281 | 0.652 | 0.371 | **0.668** |
| RAA | -0.029 | 0.655 | **-0.060** | 0.630 | 0.225 | 0.657 |
| RAA + Ours | **0.000** | 0.650 | -0.082 | 0.643 | **0.090** | 0.616 |
| NRAA | 0.268 | 0.665 | 0.168 | 0.652 | 0.501 | 0.665 |
| NRAA + Ours | **0.005** | 0.650 | **0.048** | **0.654** | **0.142** | 0.657 |

potentially the first defense against such attacks as it improves fairness significantly without sacrificing utility. Also note that since the defender does not know how many points the attacker poisons, the goal here is not to discover poisoned samples, but maximizing performance on the test set compared to both pre-attack and post-attack performance by trimming a small number of training samples.

## 5.3 Defending Against Adaptive Evasion Adversaries

In addition to fairness shifts and poisoning defense, we show how the influence-based trimming approach can help defend against an *adaptive adversary* (Tramer et al., 2020; Biggio et al., 2013) that conducts evasion attacks on the given learning model. Here, adaptive refers to the adversary being able to target the given defense. In our case, the attacker can randomly select test samples to conduct the attack. Notably, this is significantly different from our robustness analysis in Section 4, where we create an adversarial set using all test samples and then trim samples to optimize robustness accuracy. Here we defend by choosing to trim the training set by a fixed amount, but we have no knowledge of what samples are adversarial during inference.

We use the attack described in Section 2 to attack logistic regression. The attacker is adaptive to randomly choose between 5-25% of the test set samples to perturb. Correspondingly, we let the defender trim only 2.5% of the training set using Algorithm 2. We repeat these experiments between the attacker and defender over 10 runs for the datasets used in Section 4. In Figure 5, we plot the pre-attack accuracy, the post-attack accuracy distribution, as well as the post-attack accuracy distribution after trimming the training set and retraining the model, on the test set. These results indicate that trimming the training set after the attack can be a viable defense. With just 2.5% of the training data trimmed, we find that we can increase performance well beyond pre-attack values.

## 5.4 Online Learning with Streaming Data and Noisy Labels

So far, we considered classification problems with a fixed training set. In this part, we consider the online classification setting with streaming batch data. Online learning is a popular choice when learning models need to be implemented on memory constrained systems, or to combat distribution shift (Hoi et al., 2021; Montiel et al., 2021). Online learning assumes that the training data arrives in sequential batches and the learning model trains/updates on each batch sequentially. Here we reuse

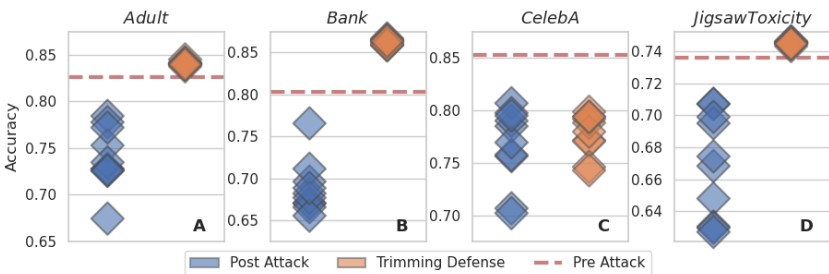

Figure 5: Defending with our trimming method against adaptive evasion attacks on four datasets.

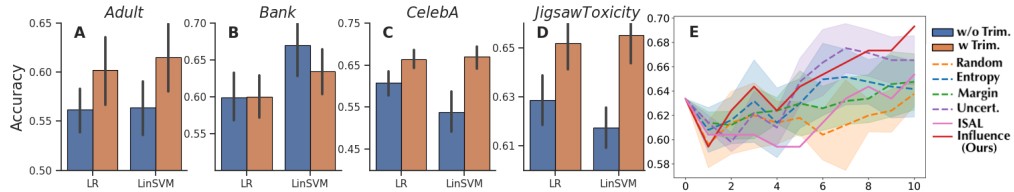

Figure 6: **A - D**: Online learning with noisy labels using logistic regression and linear Support Vector Machines with and without our influence-based trimming strategy on four datasets; **E**: Comparison of different active learning strategies on the *Diabetic Retinopathy* dataset.

the datasets in Section 4 and trim samples in each batched stream of data using Algorithm 2. Note that we estimate influence over each batch independently. To make the setting more practical, we consider that the data stream might consist of noisy class labels– where one-third of samples that arrive in a batch have noisy labels flipped from the original ground truth labels. We set the trimming budget $b$ to be 10% of the batch size, and train on 10 sequential batches of data. We then measure test set accuracy with and without trimming each batch for both the online models. Figure 6 A-D show the performance of logistic regression and linear Support Vector Machines (SVMs) (Joachims, 2006) trained in an online manner with Stochastic Gradient Descent (Ketkar, 2017) with and without our trimming processing. It can be seen that trimming combined with our influence estimation model can lead to much better performance for all the given datasets (except *Bank*) and learning models.

### 5.5 ACTIVE LEARNING

Finally, we employ our approaches for active learning by choosing beneficial samples for annotation. Active learning (Cohn et al., 1996) constitutes a small labeled training set, and a large pool of unlabeled samples, where the goal is to pick samples from the unlabeled set for annotation and then retrain the model on the combined data for boosting performance. It is generally employed in fields where annotation is very expensive to undertake, such as in medicine and health.

Here, we use a linear SVM as our learning model since a number of active learning strategies work optimally for SVM models (Kremer et al., 2014). We simulate an active learning use-case, following prior work (Zhang & An, 2016; Ahsan et al., 2022), by using the *Diabetic Retinopathy* (Decencière et al., 2014) dataset. The dataset consists of 1151 retina images of patients and can be used for predicting whether they suffer from diabetic retinopathy or not. We use the image features as extracted in (Antal & Hajdu, 2014). In this setting, there are 10 rounds of querying for annotations, where each round allows for 10 samples to be annotated. As a result, by the end of the final round we will have an additional 100 labeled samples available. We consider the following baselines for active learning– random sampling, entropy sampling (Holub et al., 2008), margin sampling (Balcan et al., 2007), uncertainty sampling (Nguyen et al., 2022), and ISAL (Liu et al., 2021) which is an influence-based strategy that uses base model predictions as pseudo-labels to compute influence. For our influence-based sampling, since we do not have access to labels in the unlabeled set, we train our tree estimation model (Algorithm 1) without labels. Then we use the estimator to predict the influence of the unlabeled samples. Out of these we select the 10 highest influence samples available in each round for labeling. Note that our influence-based sampling is deterministic over multiple runs since our tree estimator is deterministic when provided the same input. We present results for this experiment in Figure 6 E over 5 runs. It can be seen that our influence sampling outperforms other baselines in most rounds, and ends up with the best performance after the final round, demonstrating its effectiveness. We obtain similar results on our four real-world datasets in Appendix G.

## 6 CONCLUSION

In this paper, we extended influence functions to assess what data improves a given convex classifier (or a surrogate for a non-convex model) with respect to utility, fairness, and robustness by interpreting the feature space. We used tree-based influence estimation models to interpret which sample features contribute positively or negatively to the model's performance. We also designed a data selection strategy to achieve performance improvements. Through extensive experiments on synthetic and real-world datasets, and diverse application settings such as poisoning/evasion attacks, distribution shift, online and active learning, we showed how our *simple* and *general* approaches for data selection can significantly boost performance and provide valuable insights for practitioners.

## 7 ETHICS STATEMENT

Our work touches upon an important problem of selecting data for a fixed model to improve its performance in terms of utility, fairness, and robustness. For applications requiring fairness and equity (Chhabra et al., 2021; 2023b; Zafar et al., 2017; Sapiezynski et al., 2019), our approach can constitute an important preprocessing step that helps reduce bias and discrimination in systems. Our approach can also help ensure systems are robust (Xu & Mannor, 2012; Feurer et al., 2015) and high-performing under diverse problem settings, allowing these models to better serve their use-cases. It is also important to note that our work consists only of positive contributions (such as regarding interpretability, improving fairness, defending against attacks, etc.), and the approaches cannot be used in a problematic way by a malicious adversary.

Throughout the paper, we show that by employing our influence-based estimation models and simple influence-based data trimming, we can augment performance in multiple diverse problem scenarios. However, our approaches have limitations that we believe can be investigated in future work. First, better interpretability mechanisms can be designed using influence that outdo the benefits of using rule sets derived from decision trees. Second, our current data trimming approach is very simple– it is heuristic. While simplicity was the current goal of our work in this paper, better approaches for data trimming or selection can be developed. Third, our approach can be utilized on deep learning models (such as transformers (Vaswani et al., 2017)) and studied in more diverse non-convex deep learning tasks and use-cases. For instance, it is important to study robustness of large language models (Chhabra et al., 2024) and deep unsupervised models (Chhabra et al., 2022a), among others not covered in this work. Future work can delve into influence selection approaches for such models.

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

APPENDIX

## A  INFLUENCE FUNCTION DEFINITION FOR EOP

Let there be a binary sensitive attribute defined as $g \in \{0, 1\}$. The equal opportunity (EOP) fairness metric can then be defined as:

$$f^{\text{EOP-fair}}(\theta, V) = \left| \mathbb{E}_V[l(x, y; \theta)|y = 1, g = 1] - \mathbb{E}_V[l(x, y; \theta)|y = 1, g = 0] \right|.$$

Now the influence function definition is:

$$\mathcal{I}^{\text{EOP-fair}}(-x_i) = \nabla_{\hat{\theta}} f^{\text{EOP-fair}}(\hat{\theta}, V)^{\top} \mathbf{H}_{\hat{\theta}}^{-1} \nabla_{\hat{\theta}} l(x_i, y_i; \hat{\theta}). \tag{4}$$

## B  INFLUENCE FUNCTION VS ACTUAL EFFECT VIA LEAVE-ONE-OUT RETRAINING ON TOY DATA

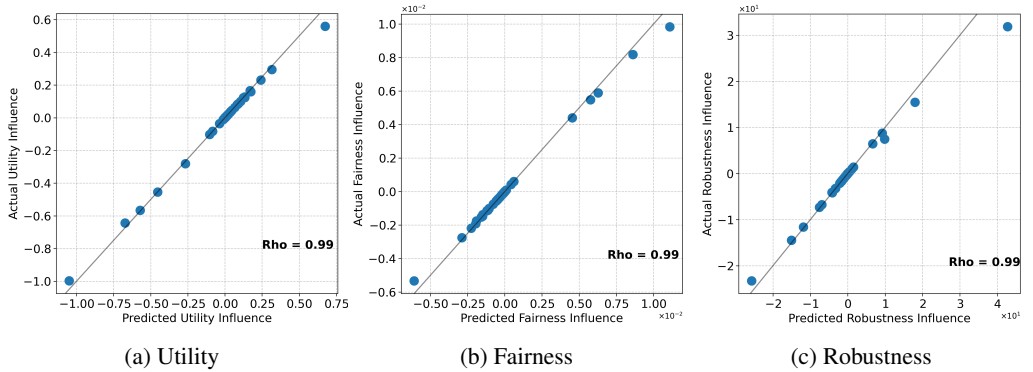

|     |     |     |
| :-: | :-: | :-: |
| (a) Utility | (b) Fairness | (c) Robustness |

Figure 7: Predicted vs actual influence analysis with leave-one-out training over 125 samples (out of 150 total training samples). Spearman's rank correlation coefficient ($\rho$) between the predicted and actual influence values is also shown.

In Figure 7 we present results comparing sample influence of points measured using influence functions to their actual influence obtained via leave-one-out retraining on the toy dataset. In the latter case we essentially measure the change in the function of interest (utility/fairness etc.) on the entire dataset versus when the point being considered is removed and the model retrained on the remaining samples. We remove a total of 125 points out of 150 samples in the training set as part of this experiment.

Figure 7a shows these results for utility, Figure 7b for fairness, and Figure 7c for robustness. It can be seen that the approximated and actual influences are very close together and we further quantify this by computing the Spearman's rank correlation coefficient $\rho$. As the value obtained in each case is 0.99, this indicates high similarity in predicted and actual influence.

## C  DETAILS REGARDING MODEL TRAINING, DATASET SPLITS, AND MISCELLANEOUS INFORMATION

### C.1  DATASET INFORMATION AND SPLITS

For *CelebA* we utilize extracted features provided by the dataset authors, and for *Jigsaw Toxicity* we extract text embeddings via the MiniLM transformer model (Wang et al., 2020). We set sex (male/female) as the sensitive attribute for *Adult*, *Bank*, *CelebA*, and ethnicity (black/other) for *Jigsaw Toxicity*. Further details are provided below.

***Bank.*** For this dataset, we have a total of 30,488 samples, with 18,292 training samples, 6,098 validation set samples, and 6098 test set samples. The number of features are 51. We use *sex* (male/female) as the sensitive attribute for fairness. The target to predict is whether or not (yes/no) a client will subscribe to a term deposit.

***Adult.*** For this dataset, we have a total of 45,222 samples, with 30,162 training samples, 7,530 validation set samples, and 7,530 test set samples. The number of features are 102. We use *sex* (male/female) as the sensitive attribute for fairness. The target to predict is whether income is $> \$50k$ (yes) or $\leq \$50k$ (no).

***CelebA.*** For this dataset, we have a total of 104,163 samples, with 62,497 training samples, 20,833 validation set samples, and 20,833 test set samples. The number of features are 39. We use *sex* (male/female) as the sensitive attribute for fairness. The target to predict is whether a given person is smiling (yes) or not (no).

***Jigsaw Toxicity.*** For this dataset, we have a total of 30,000 samples, with 18,000 training samples, 6,000 validation set samples, and 6,000 test set samples. The number of features are 385. We use *ethnicity* (black/other) as the sensitive attribute for fairness. The target to predict is whether a given tweet is toxic (yes) or not (no).

### C.2 MITIGATING DISTRIBUTION SHIFT EXPERIMENT DETAILS

In these experiments for the main paper, we basically use logistic regression as the base learning model and utilize our trimming algorithm. This constitutes the results on the vanilla model shown in the figures for each distribution shift. Now, we utilize the same points identified for deletion using our approach and measure performance before and after trimming these for all other fairness intervention baselines. Note that we use Logistic Regression as a surrogate in this case which allows us to effectively compute sample influence and improve the fairness performance of these approaches. Note that since we are using logistic regression, there is very little variance observed in the results. All the other baselines are used as provided by the authors.

### C.3 ONLINE LEARNING WITH NOISY LABELS EXPERIMENT DETAILS

Here for both models we compute influence using the logistic regression model as the surrogate. For each batch of streaming data we train a new logistic regression model from scratch and compute sample influences. Based on these values we conduct data batch stream-level trimming.

### C.4 ACTIVE LEARNING EXPERIMENT DETAILS

Here, the dataset splits for the labeled sample set, and the large unlabeled pool sample set (as well as validation/test set) are obtained by randomly partitioning the data. We then measure the results on the fixed test set and compute the influence on the fixed validation set over all rounds. For influence-based sampling, we use logistic regression as the surrogate for influence computation.

## D ADDITIONAL EXPERIMENTS FOR ALGORITHMIC PERFORMANCE ON REAL-WORLD DATASETS

In this section, we provide additional experiments that extend the results on showcasing algorithmic performance of our proposed approaches on four real-world datasets. In particular, we first provide results in a similar fashion to the main paper when the base learning model is an MLP Neural Network as opposed to a logistic regression Model. Next, we provide results for utilizing EOP instead of DP for fairness trimming experiments, for both the MLP and logistic regression models. We find that in both cases the trends demonstrate that trimming is highly beneficial and improves model performance significantly.

### D.1 RESULTS FOR MLP NEURAL NETWORK

We present results for MLP (2 hidden layers with ReLU activations in between layers) as the learning model in Figure 8. It can be seen that even though the MLP model is non-linear, our approach is

able to improve utility, fairness (DP), and robustness performance, for each of the four datasets. Furthermore, the random baseline is unable to do so, and actually leads to worsened performance in some cases.

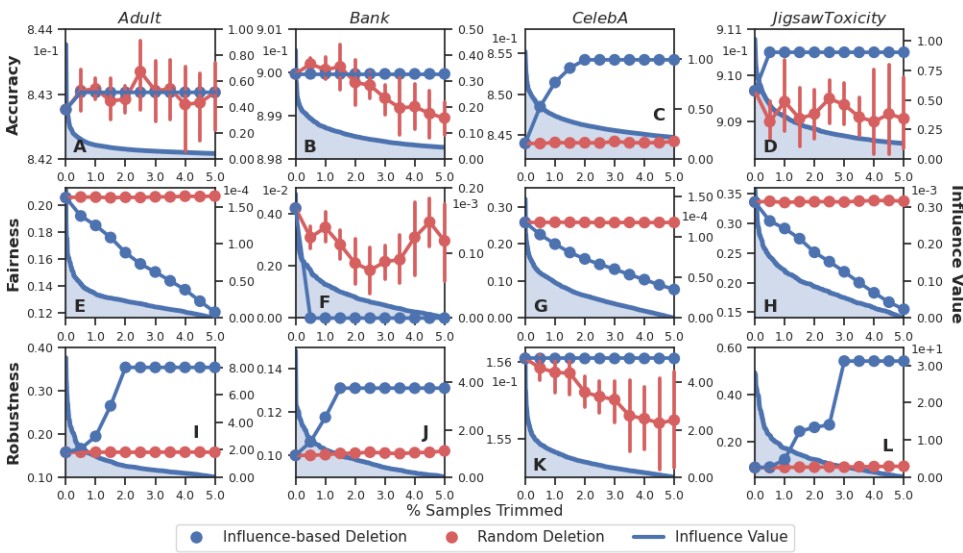

Figure 8: Double y-plot of our trimming method on the test sets of *Adult*, *Bank*, *CelebA*, and *Jigsaw Toxicity* for the MLP Neural Network as the base learning model. The left y-axes present algorithmic accuracy, fairness (DP), and robustness, while the right y-axes present the influence value of the trimmed samples. Higher values indicate better accuracy and robustness, while lower values indicate better fairness. The blue lines represent the performance of our trimming method, while the red lines represent the performance with random trimming.

## D.2 RESULTS FOR EOP (LOGISTIC REGRESSION AND MLP)

Now we present results for using EOP (instead of DP) as the fairness function for both MLP and logistic regression as the learning models as Figure 9. It can be seen that our results hold for EOP as well, and can significantly improve fairness for both models on all four datasets and for both logistic regression and the MLP network.

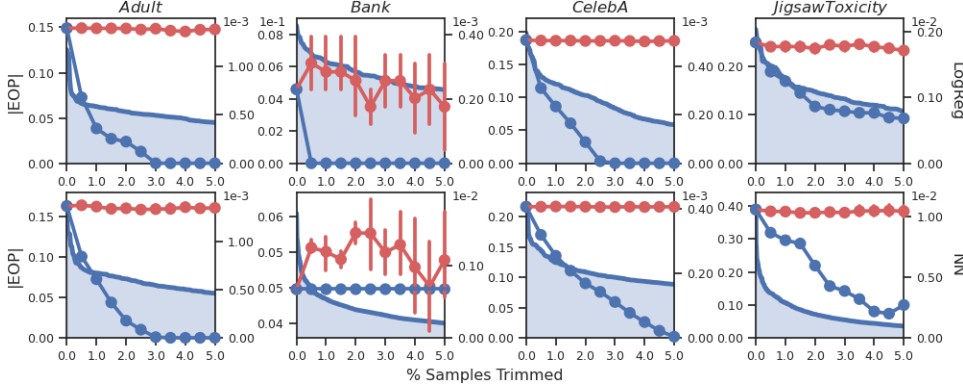

Figure 9: Additional fairness experiment on our trimming method and random trimming shown as a double y-plot on the test sets of *Adult*, *Bank*, *CelebA*, and *Jigsaw Toxicity* for Logistic Regression and MLP as the base learning model but with the EOP metric instead of DP. The blue lines represent the performance of our trimming method, while the red lines represent the performance with random trimming.

## E    Comparative Experiments with Shapley-value based Data Valuation

### E.1    Execution-time Comparisons with Influence Estimation

We now present results to demonstrate that compared to data valuation approaches such as TMC-Shapley (Ghorbani & Zou, 2019), influence is much faster, and hence, a better choice for data valuation and data selection tasks. For system details and the runtime environment, please refer to Appendix I. We utilize the implementation of DataShapley (Truncated Monte Carlo), i.e. TMC-Shapley as provided by the authors and run it with the default parameters on our four real-world datasets. We find that while influence computations are in the order of seconds ($< 3$) for all datasets except *Jigsaw Toxicity*, for which the computation takes $\sim 1.5$ minutes. This is understandable as *Jigsaw Toxicity* has the largest number of dimensions compared to the other datasets with 384 embedding features.

However, in comparison to DataShapley (TMC) these computations are many order of magnitudes faster, as that takes over 11.2 hours to compute for *Bank*, our smallest dataset. For all other datasets, we either ran into memory overflow errors after 96 hours, or the computation was not making reasonable progress (such as for *Jigsaw Toxicity*). It is clearly evident that DataShapley (TMC) is not viable for use on large-scale datasets.

Table 2: Execution times on all datasets for computation for both our influence-based approach and TMC-Shapley (Ghorbani & Zou, 2019). Influence computation is order of magnitudes faster.

| Dataset | Influence (Ours) | DataShapley (TMC) (Ghorbani & Zou, 2019) |
|---|---|---|
| *Bank* | **1.659 Seconds** | 11.2 Hours |
| *Adult* | **4.061 Seconds** | $> 96$ Hours |
| *CelebA* | **2.301 Seconds** | $> 96$ Hours |
| *Jigsaw Toxicity* | **85.42 Seconds** | $> 96$ Hours |

### E.2    Performance Comparison of Influence-Based Trimming (Ours) vs TMC-Shapley Based Trimming on Subsampled *Bank*

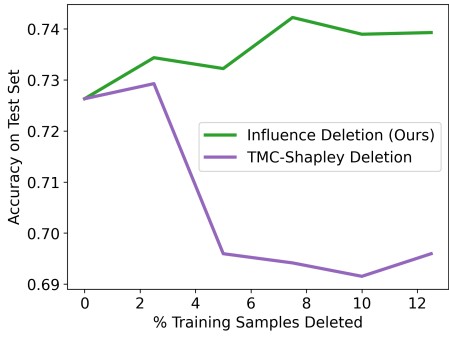

Figure 10: Comparing TMC-Shapley and Influence Estimation on a subsampled version of the *Bank* dataset, which only contains 100 samples.

Despite the running-time issues of TMC-Shapley, we conduct an additional experiment to compare performance with TMC-Shapley on a subsampled version of the Bank dataset consisting of 100 samples. We then iteratively trim the training set (up to 12.5%) using Algorithm 2 for both influence and Shapley values on logistic regression. These results are provided as Figure 10. It can be seen that influence-based trimming outperforms TMC-Shapley. While not an exhaustive comparison, we believe our approach shows promise as a data selection strategy with respect to scalability and performance.

## F    Visualizing Influence Trees for Real-World Datasets

We now showcase the visualizations of trees obtained using our first algorithm for influence estimation. For *Adult*, we provide the utility estimation tree as Figure 11, fairness estimation tree as Figure 12,

and robustness estimation tree as Figure 13. For *Bank*, we provide the utility estimation tree as Figure 14, fairness estimation tree as Figure 15, and robustness estimation tree as Figure 16. For *CelebA*, we provide the utility estimation tree as Figure 17, fairness estimation tree as Figure 18, and robustness estimation tree as Figure 19. For *Jigsaw Toxicity*, we provide the utility estimation tree as Figure 20, fairness estimation tree as Figure 21, and robustness estimation tree as Figure 22.

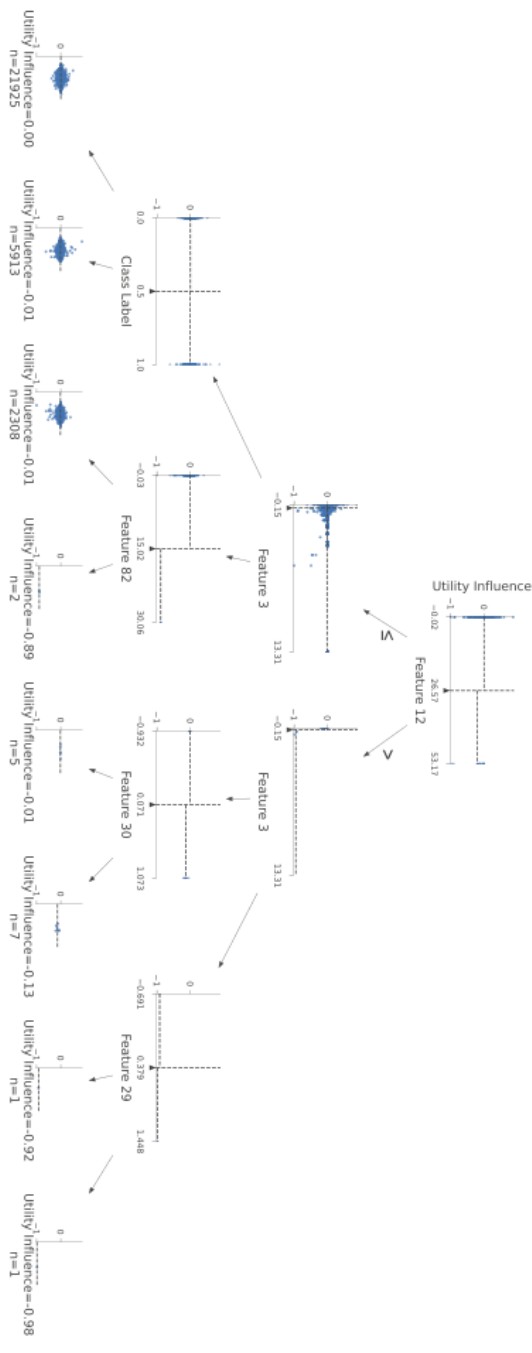

Figure 11: *Adult* dataset utility influence estimation tree.

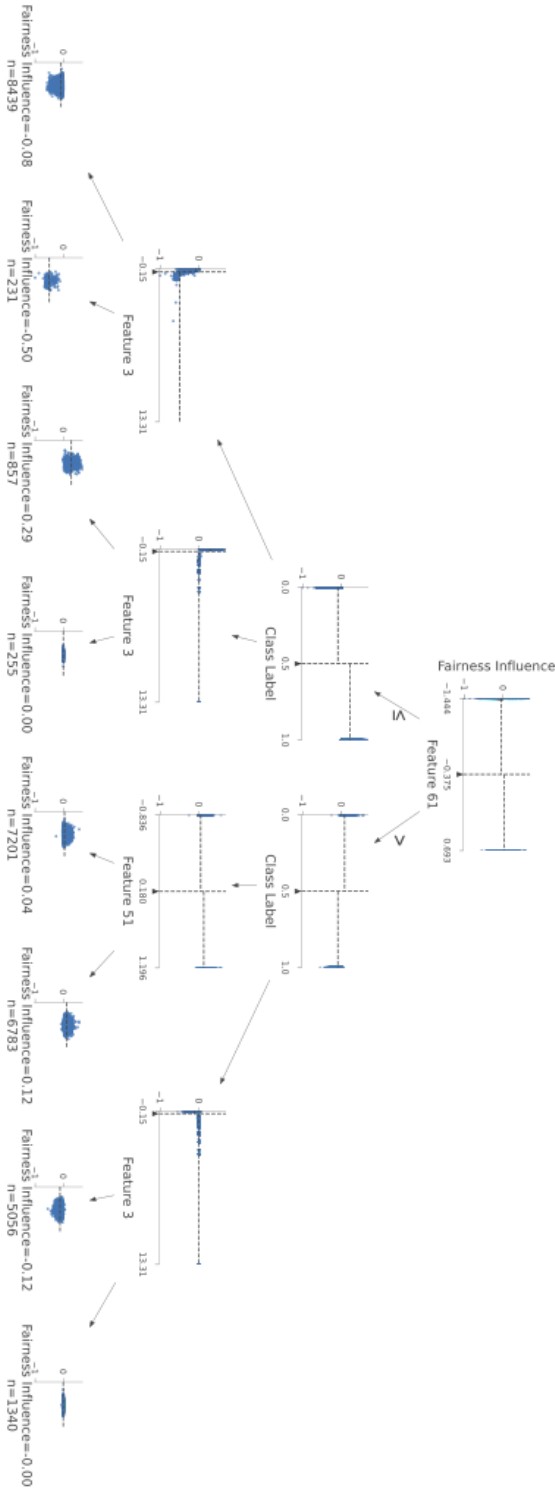

Figure 12: *Adult* dataset fairness influence estimation tree.

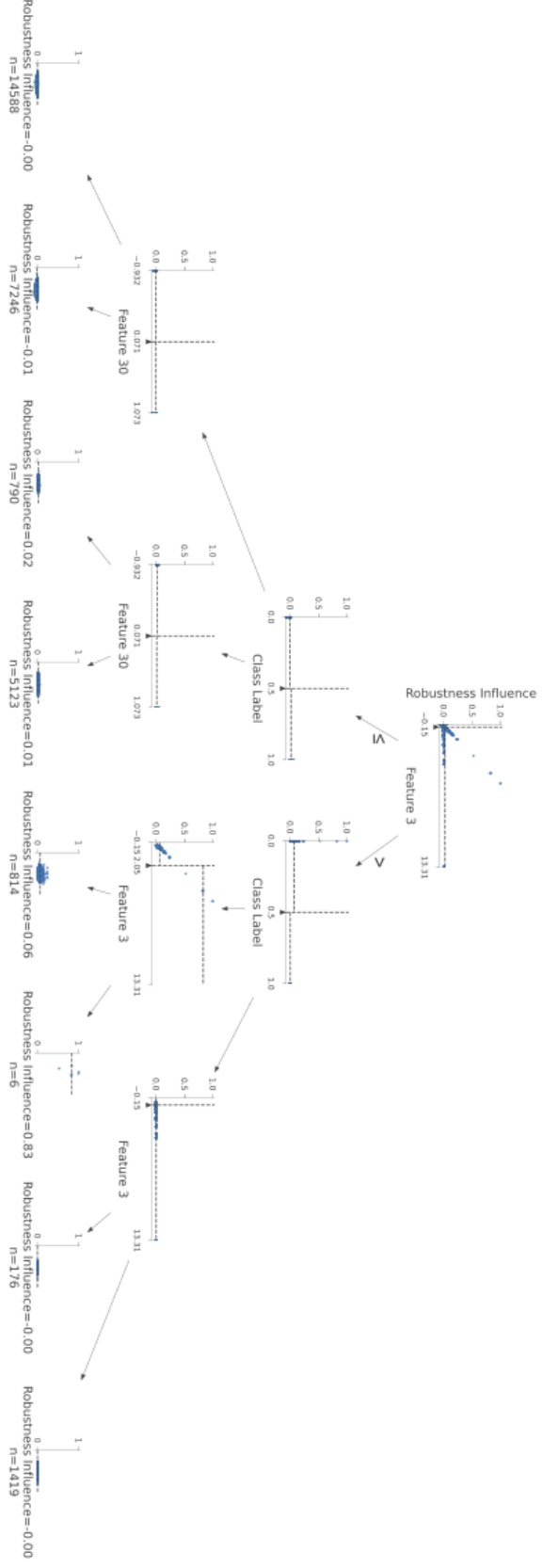

Figure 13: *Adult* dataset robustness influence estimation tree.

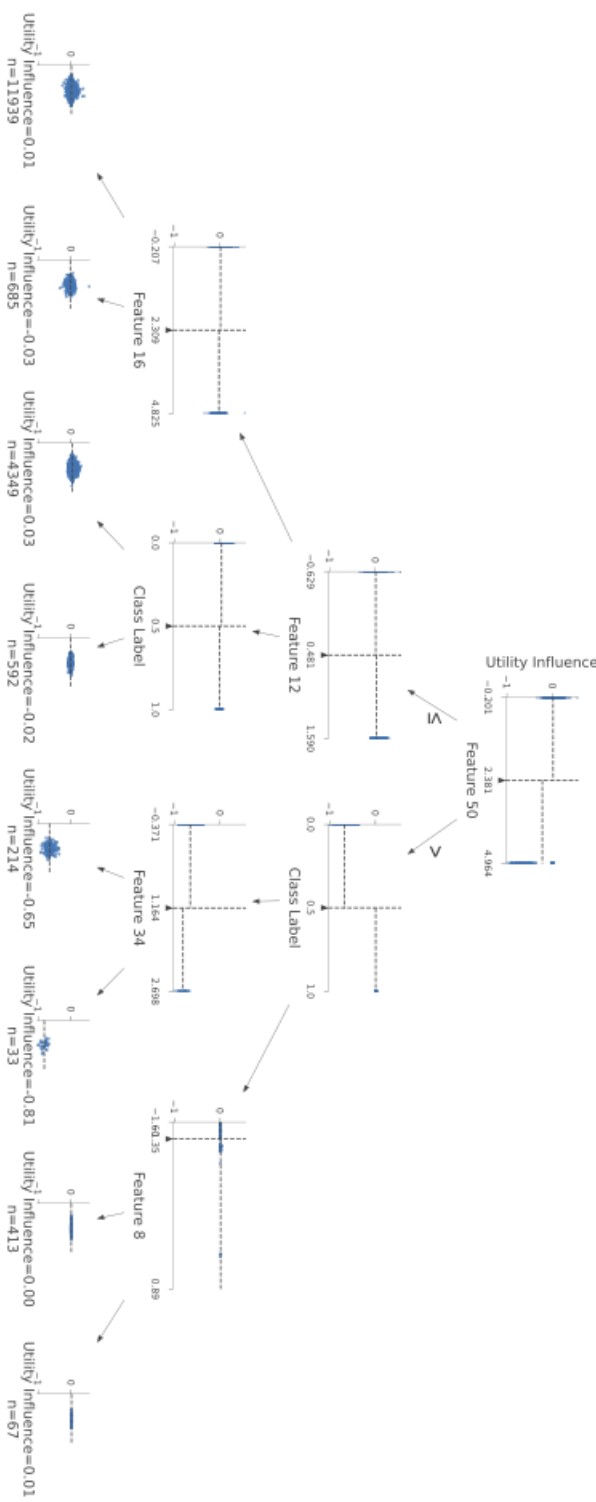

Figure 14: *Bank* dataset utility influence estimation tree.

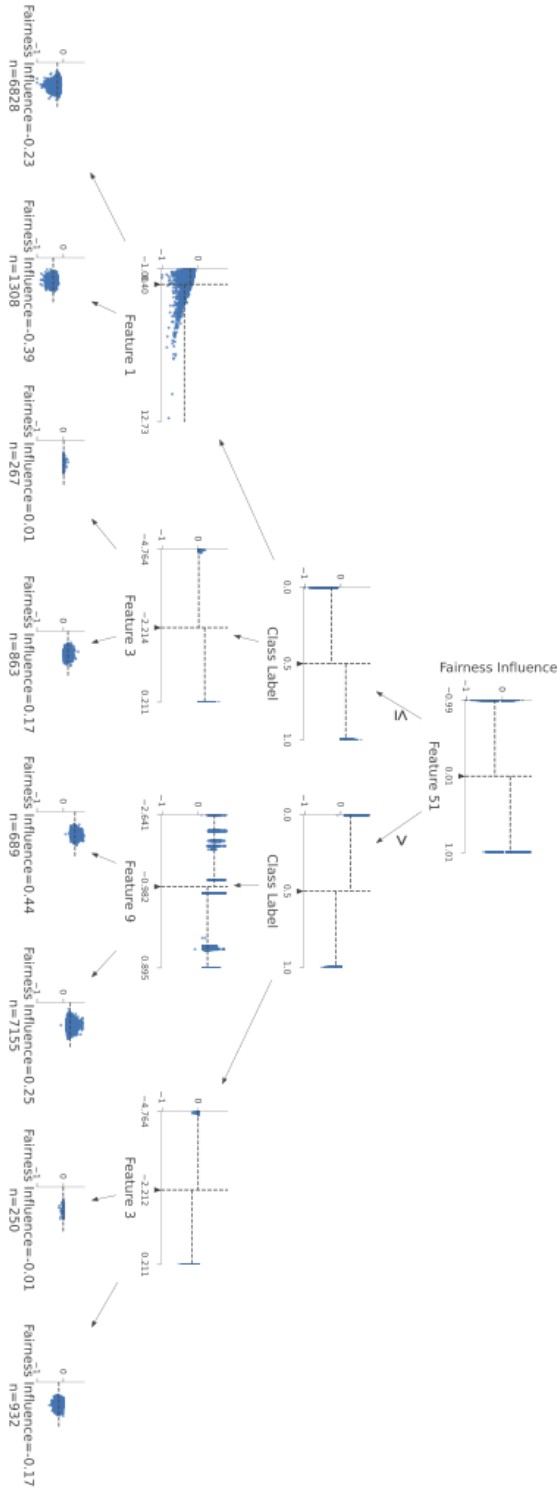

Figure 15: *Bank* dataset fairness influence estimation tree.

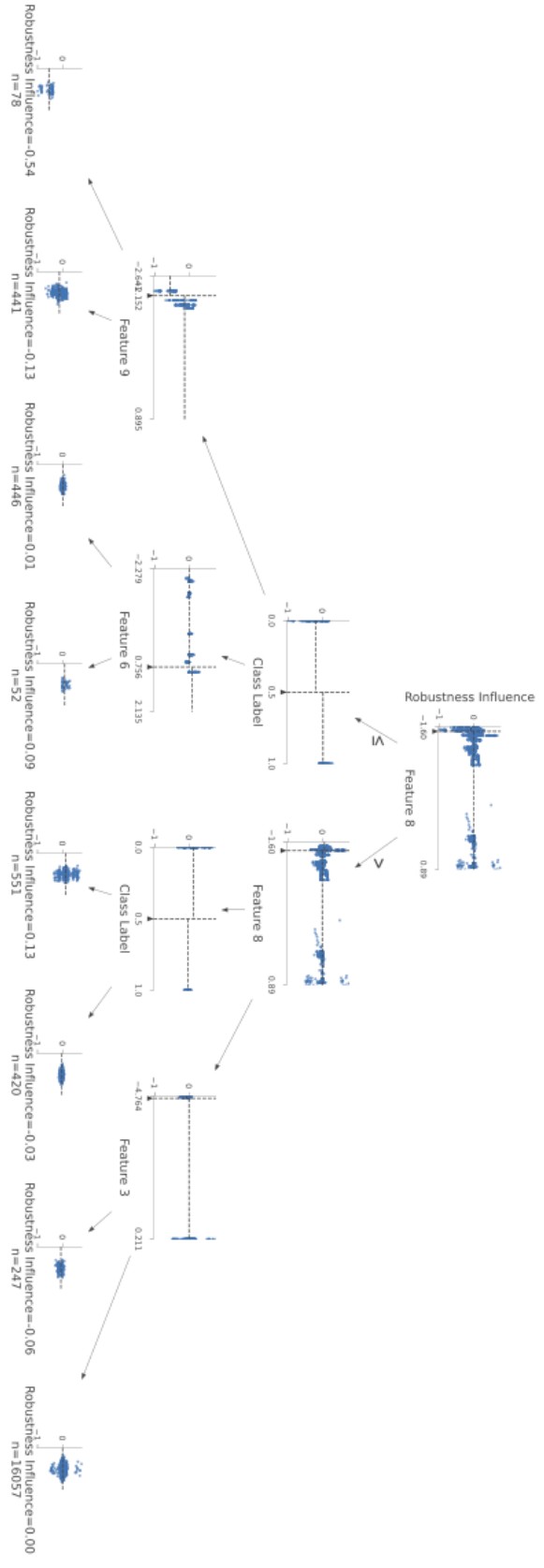

Figure 16: *Bank* dataset robustness influence estimation tree.

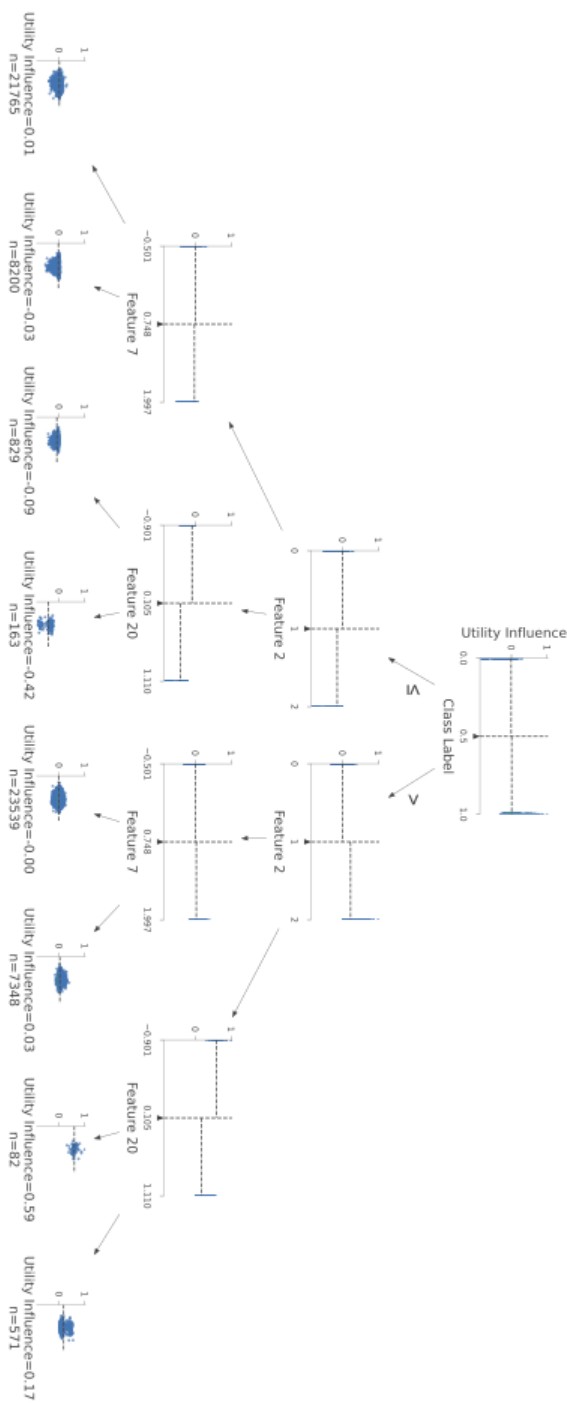

Figure 17: *CelebA* dataset utility influence estimation tree.

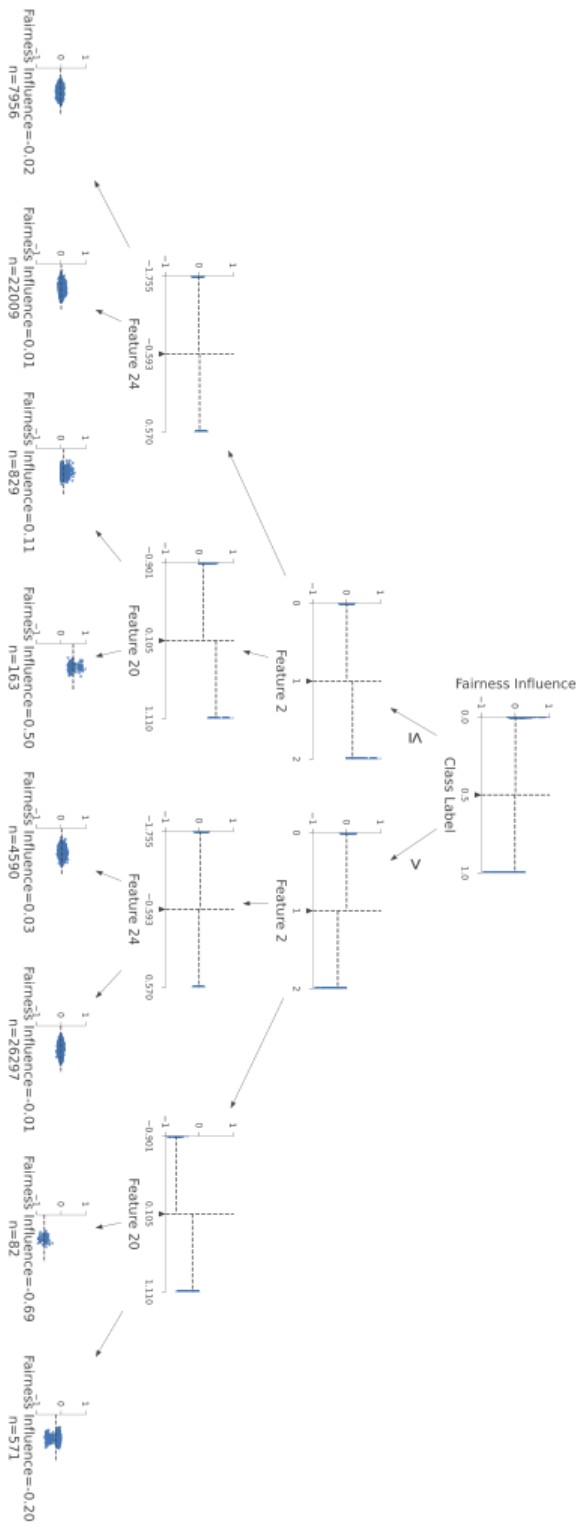

Figure 18: *CelebA* dataset fairness influence estimation tree.

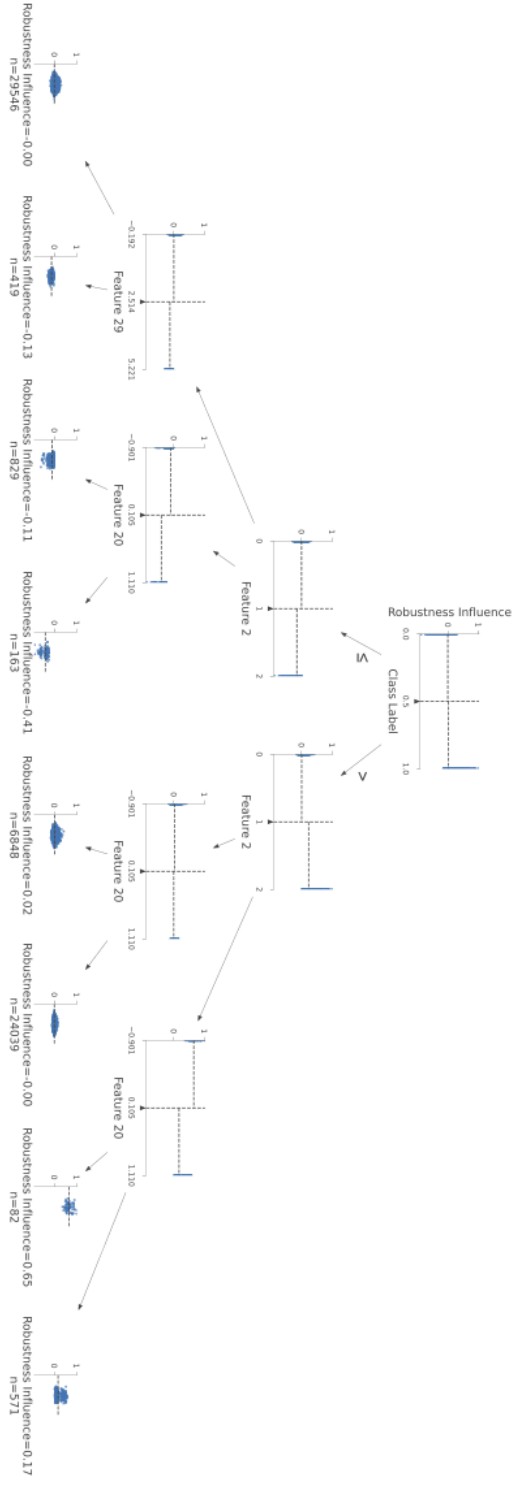

Figure 19: *CelebA* dataset robustness influence estimation tree.

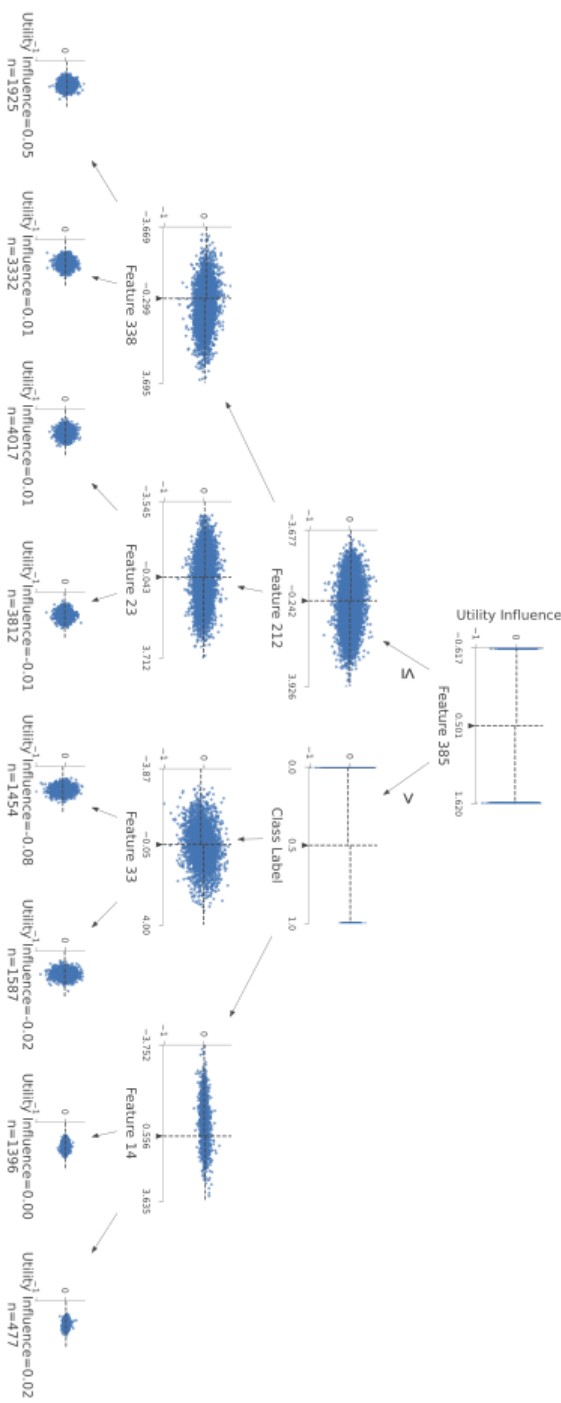

Figure 20: *Jigsaw Toxicity* dataset utility influence estimation tree.

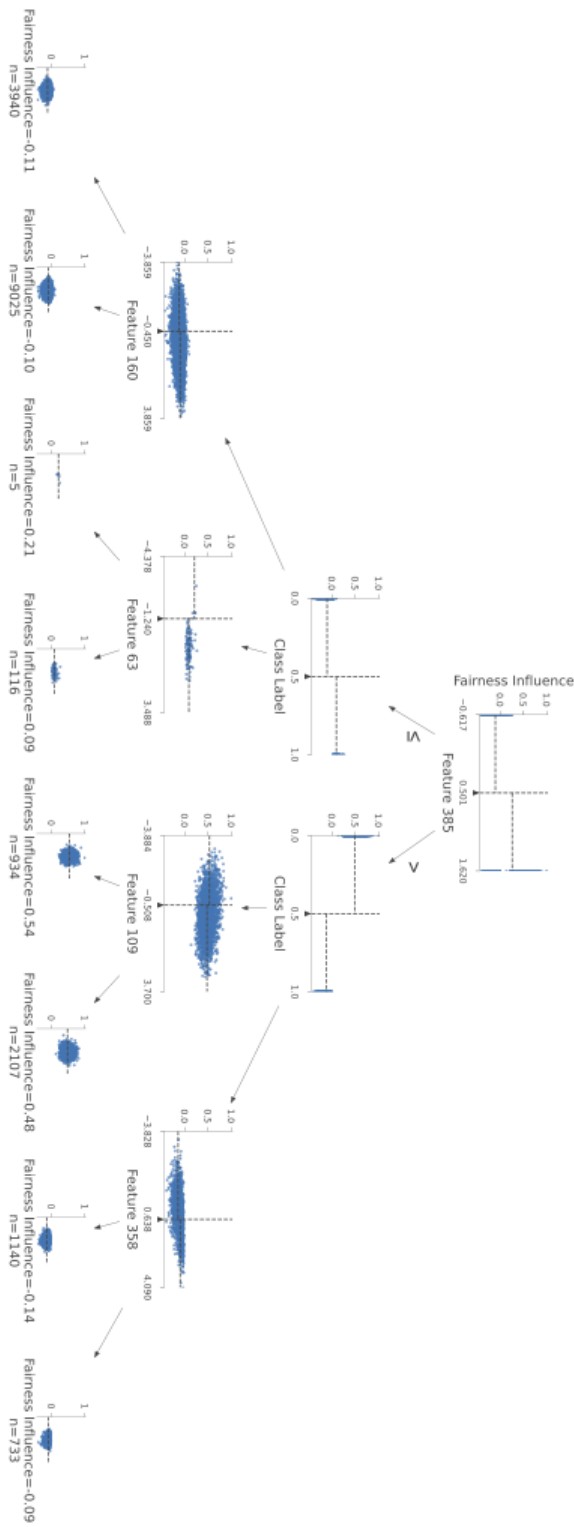

Figure 21: *Jigsaw Toxicity* dataset fairness influence estimation tree.

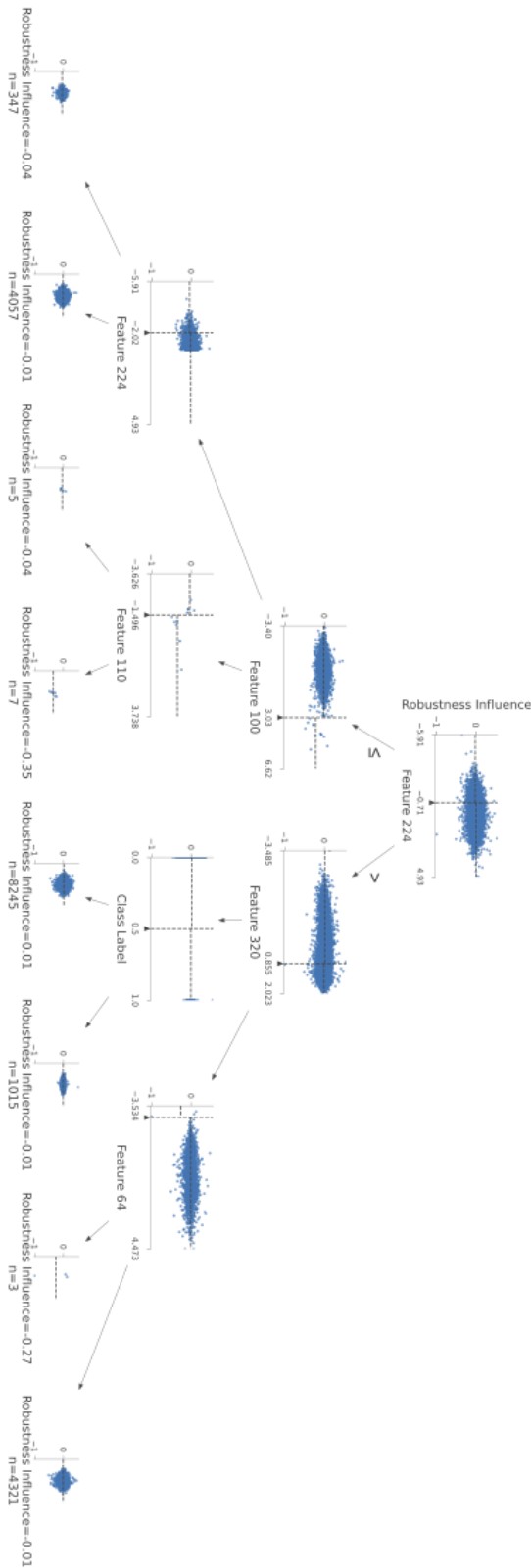

Figure 22: *Jigsaw Toxicity* dataset robustness influence estimation tree.

# G  ADDITIONAL ACTIVE LEARNING EXPERIMENTS

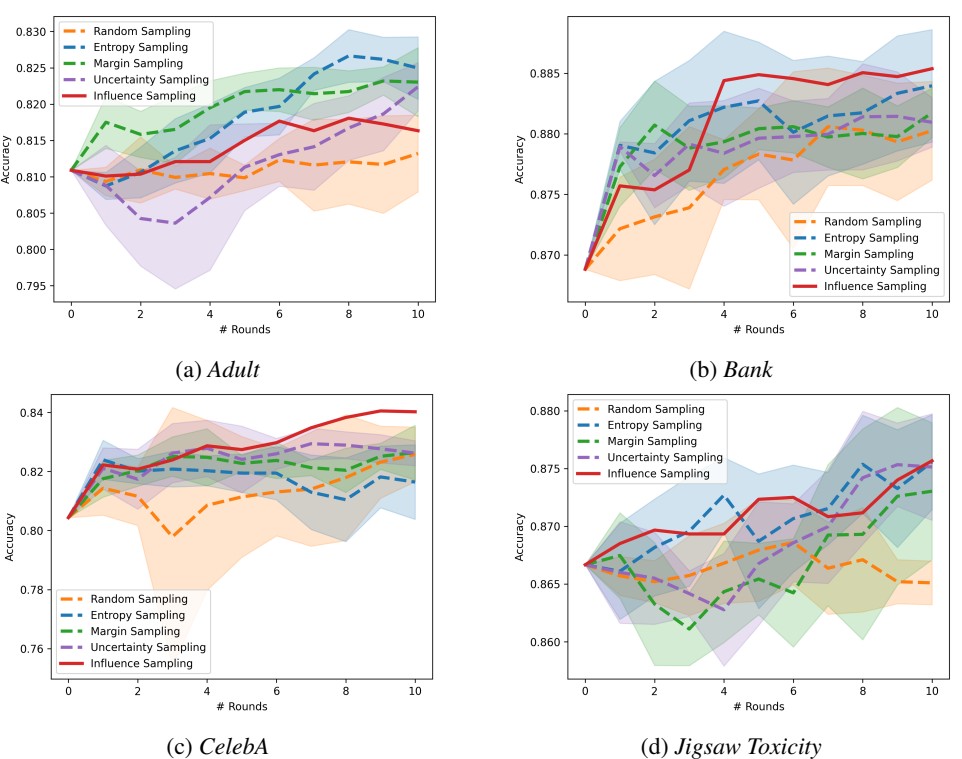

Figure 23: Additional comparisons on real-world datasets for active learning baselines with influence-based sampling.

We now provide additional results for the active learning experiments from the main paper. Here, we conduct the same experiments on our four real-world datasets. Labeled and unlabeled sample sets are partitioned randomly, as are the validation and test sets. The results are shown in Figure 23. It can be seen that influence-based sampling is superlative to the other active baselines for all datasets except *Adult*. For *Adult*, influence-based sampling still outperforms uncertainty sampling and random sampling, but is slightly less performant than entropy and margin sampling.

# H  PRELIMINARY RESULTS ON DATA TRIMMING (ALGORITHM 2) FOR IMPROVING UTILITY PERFORMANCE OF BERT

We conduct very preliminary experiments to show that influence functions and influence-based trimming can be applicable for deep learning; this can also guide future work in this area. We use the setting of (Han et al., 2020) where BERT is fine-tuned on 10k samples for binary sentiment classification on the SST-2 dataset (Socher et al, 2013) and influences are computed for individual test samples. Then we compute training set influence for 50 test samples, take the union of all training samples that were given a negative influence, and use these to trim the training set up to 12%. We fine-tune BERT for each of these trimmed sets, and measure performance. These experiments are repeated three times and results are provided in Figure 24. It can be seen that test set accuracy improves upon trimming, which shows the effectiveness of our work in an initial preliminary experiment setting.

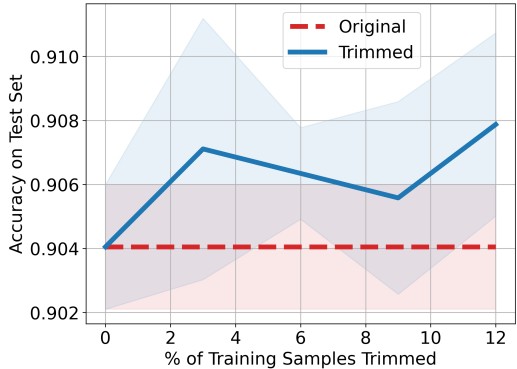

Figure 24: Comparing Influence-Based Trimming on BERT (Vaswani et al., 2017) using the experimental set-up of (Han et al., 2020) for Binary Sentiment Classification on the SST-2 Dataset (Socher et al, 2013).

## I    CODE AND REPRODUCIBILITY

• We open-source our code and experiment implementations in the following repository: `https://github.com/anshuman23/InfDataSel`.

• All experiments were conducted on a Linux server running Ubuntu 18.04.6 LTS. The CPU used was an Intel(R) Xeon(R) Silver 4114 CPU @ 2.20GHz. Any experiments using the GPU (such as for MLP, etc.) were conducted using a Tesla V100 with 32GB VRAM and CUDA version 11.4.

• All code is written in Python, and utilizes basic libraries such as as NumPy, scikit-learn, PyTorch, Pandas, etc. Detailed package information is provided in the code repository itself.

