# OpenReview forum: ""What Data Benefits My Classifier?" Enhancing Model Performance and Interpretability through Influence-Based Data Selection"
_ICLR.cc/2024/Conference — ICLR 2024 oral_

### Official Review · Reviewer_ZqCd · 2023-10-14

**Soundness:** 3 good
**Presentation:** 2 fair
**Contribution:** 3 good
**Rating:** 6
**Confidence:** 4

**Summary:**

This paper diverges from the mainstream focus on enhancing model architectures and learning algorithms on fixed datasets. Instead, it tackles an essential yet overlooked issue: understanding how a fixed convex learning model (or a convex surrogate for a non-convex model) benefits from data by interpreting the feature space. Specifically, this paper proposes to use influence estimation models to interpret the classifier's performance through the lens of data features. Furthermore, it introduces data selection methods based on influence to enhance model utility, fairness, and robustness. Through extensive experiments on both synthetic and real-world datasets, the effectiveness of the proposed method is validated. Additionally, the method proves effective not only in conventional classification scenarios but also in more challenging situations, such as distribution shifts, fairness poisoning attacks, utility evasion attacks, online learning, and active learning.

**Strengths:**

- The research topic is realistic and important. In the era of big data, the analysis of "more important data points" is significant.
- Experimental results are great. In a series of tasks, the proposed method can achieve the best performance.

**Weaknesses:**

- The motivation of this paper is not clear and not strong.
- Technical contributions of the proposed method are limited.
- Writing is not unsatisfactory. Many times, readers are unable to understand the author’s true intentions.

More details about the weaknesses can be checked below.

**Questions:**

- At the beginning, this paper claims it is related to data valuation,  data influence, and data efficiency. Essentially, this paper studies the problem of "coreset selection", which is not a new problem in machine learning. Coreset selection surely is related to the above topics. Therefore, it seems that there is no need to introduce so much redundant content in the main paper.
- The motivation is not clear. It has been fully studied to use the influence function to analyze the importance of data points. This paper follows this line. However, after checking this paper, I am confused about the proposed method of this paper, as the paper just combines the influence estimation and decision tree. Also, why do we need this tree?
- This paper uses a lot of space to introduce the previous versions of influence functions (Section 2). However, it is not clear that the difference between previous work and this work mathematically.
- Could the paper provide more high-level intuitions about the formulas of the overall regression tree prediction and hierarchical shrinkage regularizes?
- For the method in Section 3.2, what is its time/space complexity?
- Figure 3 and the illustrations in the appendix are not informative. Could the paper supplement more descriptions for them?
- Could the paper discuss the difference between this paper and the work [1]?

----
[1] Shuo Yang et al. Dataset Pruning: Reducing Training Data by Examining Generalization Influence. ICLR 2023.

---

> ### Author Response · Authors · 2023-11-17
> **Response to Reviewer ZqCd [1/3]**
>
> We would like to sincerely thank the reviewer for their review, we are appreciative of their efforts, time, and consideration. Below we provide answers to the concerns raised:
>
> * _**Essentially, this paper studies the problem of "coreset selection", which is not a new problem in machine learning. Coreset selection surely is related to the above topics. Therefore, it seems that there is no need to introduce so much redundant content in the main paper.**_
>
>     We would like to emphasize here that coreset selection and our work differ in multiple ways and our main motivation is not aligned with the motivation for coreset selection. We provide more details on these differences below:
>
>     **Motivation and Research Question**: Coreset selection [1,2] is primarily associated with condensing and reducing the size of the training dataset for accelerating training and making training more efficient (for e.g. in deep learning) without sacrificing performance. In contrast, our work is motivated by assessing what type of data is generally beneficial/detrimental for the model by _interpreting the feature space_, and then using _data trimming/selection_ approaches to minimally modify the training set so as to improve performance with respect to _multiple functions of interest_. In our paper currently we had already cited both [1] and [2] (coreset selection approaches) and discussed them at a high-level, but we can describe these differences of motivation in comparison with our work in more detail in the revision.
>
>     **Other Empirical Justification**: To further reinforce the differences in motivation, it can be seen that for coreset selection [1,2] the training data is reduced to be just about 20-30% of the original training dataset size (that is trimming of up to 80%). This is in stark contrast to our work, where we are minimally trimming the training set-- for example, in Figure 2 we only trim up to 5%. Note that we trim minimally in all experiments of our paper, and do not consider removing a large set of samples, as data efficiency is not our main concern.
>
>     **Differences in Generality**: Different coreset selection methods generally need to be proposed for each function of interest, as it is non-trivial to apply a coreset selection method for accuracy to fairness [3]. This is also in stark contrast to our work, where we seek to have general and simple approaches that can easily be applied to multiple functions of interest (fairness, accuracy, robustness, etc.) in a trivial way. This makes our approaches more general than coreset selection, which would be more specific. This is also observable by the widespread application of our methods to problem scenarios that extend beyond conventional classification, such as online learning and active learning. For instance, it is not immediately clear how coreset selection could be applied to active learning.
>
>     ___
>
> * _**Could the paper provide more high-level intuitions about the formulas of the overall regression tree prediction and hierarchical shrinkage regularizes?**_
>
>     We apologize for any lack of clarity and will make these descriptions clearer in the main text.
>
>     **Overall Regression Tree Prediction**: Since the predictions for a particular node are dependent on where it lies in the tree, the tree model prediction can be denoted as a telescoping sum. We start with the average prediction response at the root node and then keep summing over the individual prediction response at each node (calculated as differences in the average prediction response between successive nodes). This provides us with the prediction at a particular query leaf in the tree.
>
>     **Hierarchical Shrinkage Regularization**: Without hierarchical shrinkage the tree might have some leaves that noisily influence estimates as they have only a few samples. This can affect both interpretability of the tree as well as predictive capability. To counteract this, hierarchical shrinkage is a powerful yet simple approach. Here, we replace the average prediction response over a leaf with a weighted average of the mean or average responses over the leaf and each of its ancestors. The weights depend on the number of samples in each leaf, and are controlled by the regularization parameter $\lambda$ specified at run-time. Mathematically, this is observable with the slight modification to the summation over individual predictive responses at nodes where we divide the predictive response by the number of samples the node contains as well as $\lambda$, the regularization parameter.
>
>     ___

---

> ### Author Response · Authors · 2023-11-17
> **Response to Reviewer ZqCd [2/3]**
>
> * _**This paper uses a lot of space to introduce the previous versions of influence functions (Section 2). However, it is not clear that the difference between previous work and this work mathematically.**_
>
>      - The difference between previous work (such as Koh and Liang's work) is that those provide an approach for computing influence at the sample-level whereas we are trying to derive general "rule sets" at the feature-level for how features affect influence. For example, in Koh and Liang's work, they can obtain influence $I(x_i)$ for a sample $x_i$ (or change in output with sample-level perturbations). In our work, we use a large set of such samples (and their labels) and their influence values, to train an interpretable estimator, such as CART. This can then be used to derive a general interpretation for influence using the feature space-- for example some rules for Accuracy from our 2-D feature space in Figure 1 are: Feature 1 $>= 1.9$ and Feature 2 $< 0.1$ lead to high influence while Feature 1 $< 1.9$ and Feature 2 $< 0.1$ lead to low influence.
>
>     - This sort of feature-level interpretation of performance gain via influence has not been undertaken in the literature previously. This can be useful in many scenarios, some of which we outline below but are not limited to only these:
>
>         **(1)** When new data needs to be added that we do not have influence scores for yet, we can use the influence estimator to obtain influence scores simply via inference. For Koh and Liang's approach, they would still need to train the model once with all samples under consideration every time a new sample arrives.
>
>         **(2)** In algorithmic data recourse scenarios [4] such as to describe why an individual was denied a loan, it can be used to provide an interpretable justification for why a decision was made simply by describing the tree rules that were either met or violated by the data sample.
>
>         **(3)** In active learning, traditional influence computation (such as using Koh and Liang's work) is not viable since there is no access to the sample labels. Previous work such as ISAL [5] (one competitive method in our paper) have still utilized influence in these scenarios, but these do not work as well, since they use the base model itself to generate pseudo-labels for prediction. Our approach using the decision tree serves as an effective and much better alternative, as is evident in the active learning experiments of our paper (Section 5.5 - Figure 6E), where we compare with ISAL.
>
>         **(4)** In some sense, we are bridging feature-interpretation approaches (such as Shapley values and LIME) with sample-interpretation approaches (such as Data Valuation and Influence), and hence, can accommodate any applications that these are used for.
>     ___
>
>
> * _**I am confused about the proposed method of this paper, as the paper just combines the influence estimation and decision tree. Also, why do we need this tree?**_
>
>     Please refer to our response above for our answer to this question. Thank you. If the reviewer raised extra concerns, we are happy to address them.
>
>     ___
>
>
> * _**For the method in Section 3.2, what is its time/space complexity?**_
>
>     Our algorithms are computationally efficient and can scale with large datasets. We provide analytical details on the time complexity below.
>
>     **Algorithm 1 Time Complexity**: For calculating computational complexity, we need to analyze the two steps of our influence estimation process. First, the influence computation of training samples and a given base model requires $\mathcal{O}(npm)$ time where $n,p$, and $m$ are number of training samples, model parameters, and test samples respectively. Please refer to Section 3 of Koh and Liang's paper for how influence can be computed in $\mathcal{O}(npm)$ using stochastic estimation or conjugate gradients. Second, the cost of training time of the influence estimation tree in the worst case is $\mathcal{O}(n^2 d)$ where $d$ is the number of features. Thus, the overall complexity is $\mathcal{O}(npm) + \mathcal{O}(n^2 d)$. Note that the quadratic dependence of the second term can be made linear in $n$ simply be using more sophisticated tree-building methods [6] which have a time complexity of $\mathcal{O}(nd)$. Since the overall time complexity will then be linear: $\mathcal{O}(n(pm + d))$, the approaches proposed are amenable to large scale datasets (CelebA in our experiments has 104K samples). Furthermore, our approaches work with very high-dimensional data as well. As shown in experiments on the image (CelebA) and NLP (Jigsaw Toxicity) datasets we consider in the paper, we utilize embeddings obtained by deep learning models as a preprocessing step prior to our approach. This significantly reduces the dataset feature size, and at the same time allows for a rich and powerful feature space that can be used to train our models/methods.

---

> > ### Author Response · Authors · 2023-11-17
> > **Response to Reviewer ZqCd [3/3]**
> >
> > * _**For the method in Section 3.2, what is its time/space complexity?**_ (Continued)
> >
> >     **Algorithm 2 Time Complexity**: The time complexity of Algorithm 2 is easier to calculate. Note that the influence values provided once at input require $\mathcal{O}(npm)$ to calculate as mentioned before. Then the sorting step requires $\mathcal{O}(n log(n))$ in the worst case. The other operations (lines 8-9) are linear vector operations, leading to a worst case time complexity of $\mathcal{O}(n(pm + log(n))$. This is not a computational bottleneck for large datasets.
> >
> >     ___
> >
> > * _**Figure 3 and the illustrations in the appendix are not informative. Could the paper supplement more descriptions for them?**_
> >
> >     We apologize for the lack of clarity and understanding. We will aim to provide more descriptions regarding these in the main text as follows:
> >
> >    _Figure 3 provides an example of an influence subtree that showcases rule sets for how samples contribute positively or negatively to the model with regards to fairness. Starting at the root node we can observe how Feature 385 (possibly related to discerning some detail of authorship of the text) contributes to the influence decisions the model makes. We only visualize the left half of this tree where this feature value is less than 0.5. Next, the class label is used to assess fairness influence. If the label is 0 (negative class: written comment is toxic) the influence estimator will again evaluate Feature 160 and then the final influence value the sample receives will be negative. If the class label is 1 (positive class: written comment is non-toxic) the influence estimator will assign positive influence to the sample but the actual range of values will be decided based on another evaluation of Feature 63. At each level of the tree, we visualize the distribution of samples at the decision split threshold, as well as their corresponding influence values. Full influence trees in Appendix F can also be understood similarly, to interpret how feature-level rules contribute to a sample positively or negatively influences the given learning model._
> >
> >     ___
> >
> > * _**Could the paper discuss the difference between this paper and the work (Yang et. al) [7]?**_
> >
> >     Thank you for referring this work by Yang et al [7]. We had cited it in our paper in Section 2 under *Extension to Non-convex Models* as it utilized influence functions in ResNets. However, it differs significantly from our work with regards to motivation. Based on the reviewer's feedback we will discuss it in the main paper in contrast with our work as follows:
> >
> >     _The goal of Yang et al differs from ours with regards to motivation. The authors seek to reduce/trim the size of the training dataset as much as possible while ensuring that model performance (utility) on the new training set is relatively similar to the original dataset. In contrast, our goal is to trim a very small percentage of the dataset, while ensuring that model performance is improved with respect to multiple functions of interest (fairness, utility, robustness)._
> >
> > ___
> > ___
> >
> > **References:**
> > 1. Krishnateja Killamsetty et al. "Retrieve: Coreset selection for efficient and robust semi-supervised learning." NeurIPS 2021.
> > 2. Baharan Mirzasoleiman et al. "Coresets for data-efficient training of machine learning models." ICML 2020.
> > 3. Huang, Lingxiao et al. "Coresets for clustering with fairness constraints." NeurIPS 2019.
> > 4. Ustun, Berk, Alexander Spangher, and Yang Liu. "Actionable recourse in linear classification." ACM FAT* 2019.
> > 5. Liu, Zhuoming, et al. "Influence selection for active learning." CVPR 2021.
> > 6. Su, Jiang, and Harry Zhang. "A fast decision tree learning algorithm." AAAI 2006.
> > 7. Shuo Yang et al. "Dataset pruning: Reducing training data by examining generalization influence." ICLR 2021.

---

> > > ### Comment · Reviewer_ZqCd · 2023-11-19
> > > **Response**
> > >
> > > Thanks for your detailed feedback that addresses my concerns properly. I thus decide to increase my score.
> > >
> > > The remained concern is that could the paper provide a comparison with previous works about time/space complexity.

---

> > > > ### Author Response · Authors · 2023-11-20
> > > > **Additional Response to Reviewer ZqCd**
> > > >
> > > > Dear Reviewer ZqCd, thank you for your prompt response, we appreciate your help in strengthening our paper.
> > > >
> > > > Regarding the additional comparisons with time complexity, we agree with the reviewer. We have provided additional details comparing time complexity of multiple approaches, categorized across sections of the paper (Section 4 in the conventional classification setting and Section 5.1 with a number of fairness intervention baselines):
> > > >
> > > > * **Influence Trimming Comparison Baselines (Section 4):**
> > > >
> > > >     * As described previously, for our methods, Algorithm 1 can achieve a time complexity of $\mathcal{O}(n(pm + d))$ (by using better tree-building methods). For Algorithm 2 the time complexity is $\mathcal{O}(n(pm + log(n)))$.
> > > >     * For a Shapley value based data valuation approach, (for example TMC-Shapley of Ghorbani et al), the time complexity will be $\mathcal{O}(2^{n}mT) + \mathcal{O}(nlog(n))$ where $T$ is the training iteration count (epochs). This complexity is computed similar to time complexity for Algorithm 2: the first term is the data valuation step, and the second is the sorting/trimming step. It can be seen that just by the first term time alone, time complexity is exponential in $n$ which is much slower than our methods.
> > > >     * Further, note we can remove the dependence of the number of parameters $p$ in our approach by using more improved influence computation approaches such as Representer Point (Yeh et al, 2018). The improved time complexity of Algorithm 1 would then be $\mathcal{O}(n(m + d))$ and of Algorithm 2 would be $\mathcal{O}(n(m + log(n)))$. This modification would lead to very high computational efficiency.
> > > >
> > > > ___
> > > >
> > > > * **Fairness Distribution Shift Methods (Section 5.1):**
> > > >
> > > >     * Influence-based Reweighing (Li et al, 2022): First, this method also requires an influence computation step similar to our work which can take $\mathcal{O}(npm)$ time. Then it solves a linear program (LP) with $n$ variables, and using interior point methods an LP can be solved in polynomial time. For e.g. using Vaidya's method (Vaidya, 1989) an LP can be solved in $\mathcal{O}(n^{2.5})$. For reference, this leads to an overall time complexity much more than our methods.
> > > >
> > > >     * Correlation Shift (Roh et al, 2023): This method requires solving a convex relation of a quadratically constrained convex program which can also be solved in (weakly) polynomial time using interior point methods, but for reference this will also be a larger time complexity than our methods.
> > > >
> > > >     * FairBatch (Roh et al, 2021): This method requires solving a bi-level optimization problem, which consists of an upper- and lower-level optimization problem, both of which are interdependent on one another. Bi-level optimization can often not be bounded in time, but can be solved in polynomial time under certain constraints (please refer to Deng et al, 1998 for more information on these problem classes).
> > > >
> > > >     * Robust Fair Logloss Classifier (Rezaei et al, 2020): This method requires training a classifier from scratch, and hence, should have significant time complexity, especially if the data under consideration has a large number of samples and is high-dimensional with a large number of features.
> > > >
> > > >
> > > > We would like to thank the reviewer again for their efforts. If they would like us to compare time complexity of any other approaches, we can also aim to do so.

---

> > > > > ### Comment · Reviewer_ZqCd · 2023-11-20
> > > > > **Response**
> > > > >
> > > > > Thanks for your answers. I become more positive about this submission.

---

> > > > > > ### Author Response · Authors · 2023-11-20
> > > > > >
> > > > > > Dear Reviewer ZqCd, thank you for engaging with us and helping us improve our paper, we appreciate it.
> > > > > >
> > > > > > Kind Regards,
> > > > > >
> > > > > > Authors.

---

### Official Review · Reviewer_MEKL · 2023-10-31

**Soundness:** 3 good
**Presentation:** 2 fair
**Contribution:** 3 good
**Rating:** 8
**Confidence:** 3

**Summary:**

The authors propose an influence-based trimming approach to uncover which samples and features contribute positively/negatively to the specified utility function. The authors perform experiments with various utility functions (fairness, accuracy, etc) on several datasets: adult, German, drugs, and celebA, among others.

**Strengths:**

- From my understanding, the authors are trying to not only get the best data samples useful for the model but also identify the best features of the data to use. They use regression trees to help in feature selection and use the influence function for the sample selection.
Neither influence estimators for sample valuation (for different utilities: fairness, accuracy, data poisoning, etc..) nor CART as a feature selector is a new concept, but the combination is a useful endeavor and an interesting perspective.

- Authors carry out several experiments on several datasets and investigate the performance of their method on several applications, and also compare their work with TMC Shapley.

**Weaknesses:**

- When I read feature space, I think of d-dimensions where the variables (features) live. The authors' writing was a bit confusing to me because from the abstract to the introduction, I thought their influence estimation-based method was identifying features from the feature space with positive/negative influence on the model utility (accuracy, fairness, robustness, and so on).
However, at the beginning of the background section and throughout the experiment results, the authors focus on only the contribution of the training samples to the utility function.
I think the authors should be a bit more clear in the writing or presentation.  Although section 3 is fairly written, I would recommend that authors revisit abstract+sections 1-3.

- Since the authors focus on features and samples, it would have been informative to see the difference in selected/excluded features and samples and the consequential contribution to the utility with and without the authors' method.

- Although influences functions are not affected by retraining-related complexity, they have a high incremental complexity due to the computation of the Hessian matrix for each x_{i} valuation, which might worsen (beyond retraining) when n is large.
Additionally, using CART as a sub-module further increases model complexity.
I would have appreciated looking at the code specific to section E.1 in the appendix (I couldn't find it in the shared code base)


- Not entirely sure, probably it's the figure, I find the almost constant utility values with random deletion somewhat unrealistic.
Could the authors also explain Figure 2C?
The scale for accuracy on some figures in 2 is not intuitive. Is it possible for authors to adopt similar scales for similar utilities across datasets?

- Experimental results.
  - Figure 2 Specific questions: I find the almost constant utility values with random deletion somewhat unrealistic.  Could the authors also explain Figure 2C?
  - Figure 10 in the appendix.  If you're removing low-value samples, I wouldn't expect TMC-Shapley to behave like that, accuracy would increase with the removal of low-value samples.  If you're trimming high-value examples, then this graph would make sense but would mean influence-based trimming is performing poorly.
  - Instead of TMC-Shapely and random, it would have been more informative to see how the proposed approach compares with other influence estimation-based approaches, including vanilla (without CART) influence estimation.
  - The scale for accuracy on some figures in 2 is not intuitive. Is it possible for authors to adopt similar scales for similar utilities across datasets?


- Minor:

  - While the focus on convex loss is understandable, it might lead to sub-optimal influence value estimation due to the model parameters not being at a stationary point or the model not converging. This might then be a net negative and misleading data value estimation.
  - It looks like the authors do one utility at a time. Due to often competing utilities,  for example, key features and samples for fairness might not necessarily be the same for accuracy, and in most cases might have a negative influence. It would be interesting to see an interplay of various utilities.
  - Although authors use several datasets, all of them are binary settings. Value computation increases with classes, so I am curious to know if this is the reason authors only focused on binary settings or if there is another reason behind this design choice.
  - The authors' paper was 32 pages instead of 9

**Questions:**

While I think the authors propose an interesting perspective, the presentation of the paper needs some improvement.
I have raised my main concerns in the weaknesses section.

---

> ### Author Response · Authors · 2023-11-17
> **Response to Reviewer MEKL [1/3]**
>
> We would like to thank the reviewer for their time, effort, and consideration in reviewing our work, we are very grateful. We provide additional details to answer questions raised:
>
> * _**I think the authors should be a bit more clear in the writing or presentation. Although section 3 is fairly written, I would recommend that authors revisit abstract+sections 1-3.**_
>
>     Thank you for the concrete suggestion for improving our paper. We will work to improve the writing and readability, and add a section on _Problem Formulation_ that clarifies the presentation of the problem.
>
>     ___
>
> * _**Since the authors focus on features and samples, it would have been informative to see the difference in selected/excluded features and samples and the consequential contribution to the utility with and without the authors' method.**_
>
>     If we are correct in understanding the reviewer's suggestion, you would like to see the difference in performance and utility when our approach is used and when it is not (that is, the performance of the original model). This is currently observable in Figure 2 for each subfigure when the x-axis value = 0 (0% samples trimmed or excluded). Note that this will be a constant value throughout as there is little variance in the output of logistic regression. If the reviewer intended something else, we can provide additional details during the discussion phase.
>
>     ___
>
>
> * _**Although influences functions are not affected by retraining-related complexity, they have a high incremental complexity due to the computation of the Hessian matrix for each xi valuation, which might worsen (beyond retraining) when n is large. Additionally, using CART as a sub-module further increases model complexity. I would have appreciated looking at the code specific to section E.1 in the appendix (I couldn't find it in the shared code base)**_
>
>     - We apologize for not including the Appendix E.1 code with the provided codebase as we primarily used the original code of TMC-Shapley (Ghorbani et al). However, we have added this experiment in the following code repository: <https://anonymous.4open.science/r/ICLR_Rebuttal_Experiments> for the reviewer to go through. This also has implementations of additional baselines which we discuss in a subsequent response. Next, we discuss analytical details on time complexity of our algorithms.
>
>     - The tree-based influence model of Algorithm 1 is computationally efficient and can scale with large datasets. It has a training time of $\mathcal{O}(npm) + \mathcal{O}(n^2 d)$ in the worst case where $n$, $p$, $m$, and $d$ are the number of training samples, the number of model parameters, the number of test samples, and the number of features, respectively. The first term is for the influence computation step (please refer to Section 3 of Koh and Liang's paper for how influence can be computed in $\mathcal{O}(npm)$ using stochastic estimation or conjugate gradients) and the second is for training and constructing the tree model. The biggest factor here is the quadratic dependence on the number of training samples, which can be made linear in $n$ simply by using more sophisticated tree-building methods [1] that have a time complexity of $\mathcal{O}(nd)$. This would result in an improved overall time complexity of $\mathcal{O}(n(pm + d))$. Thus the model can scale easily to larger datasets on modern hardware. This is also evident in our experiments in the paper on the CelebA dataset that has 104K samples and runs computationally efficiently even with a standard tree-building algorithm such as CART (it takes 2.455 seconds to run on average on a Linux server with an Intel Xeon 2.2GHz CPU).
>
>     ___
>
> * _**The scale for accuracy on some figures in 2 is not intuitive. Is it possible for authors to adopt similar scales for similar utilities across datasets?**_
>
>     We understand the reviewer's concern regarding the figure scales; unfortunately, employing same scales across the different subfigures for a function of interest would make it challenging to observe the differences in the trends across different datasets. For example consider accuracy-- for Adult the values vary from 0.826 to 0.832, for Bank they vary from 0.80 to 0.90, and for Jigsaw Toxicity they vary from 0.735 to 0.744. Thus, using the same scale (say 0.735 to 0.90) would make all the trends for datasets except Bank look like a constant line. This is because the metric values are highly dataset-dependent and comparative analysis necessitates individual scales for each dataset.
>
>
>     ___

---

> > ### Author Response · Authors · 2023-11-17
> > **Response to Reviewer MEKL [2/3]**
> >
> > * _**Figure 2 Specific questions: I find the almost constant utility values with random deletion somewhat unrealistic. Could the authors also explain Figure 2C?**_
> >
> >     - We understand the reviewer's point of view regarding constant utility values, but this is not as surprising as the very small number of randomly sampled points that are then deleted (less than 5%) might not necessarily be contributing to utility in a significant way. This trend is also observable in other work on data valuation. For example in the Data Shapley paper by Ghorbani et al, observe Figure 1 (a) and (b) and only view the scale till 5% (as the authors there consider up to 40% removals). It is evident that random deletion does not change the results that much up until a very large portion of the data is removed.
> >
> >     - Regarding Figure 2C, since CelebA is a high-dimensional complex image dataset and we are using a logistic regression model the accuracy value obtained is most likely an upper bound and the model cannot improve upon it further. That is why even after removing low influence samples, we do not see performance improvements (but it also does not drop). With random deletion, since the scale is fairly zoomed in between 0.853 and 0.854 the variance is possibly leading to more of an observable trend (it drops below the original value as well at times). Note that when we use the MLP model on CelebA which has a non-linear decision boundary we can actually improve upon accuracy much more significantly (as is observable in Figure 8C in the Appendix), backing the above hypothesis.
> >
> >     ___
> >
> > * _**Figure 10 in the appendix. If you're removing low-value samples, I wouldn't expect TMC-Shapley to behave like that, accuracy would increase with the removal of low-value samples. If you're trimming high-value examples, then this graph would make sense but would mean influence-based trimming is performing poorly.**_
> >
> >     We understand the reviewer's concern. We are removing low-influence samples. We had used the original code provided by Ghorbani et al for TMC-Shapley and obtained the trends shown. However, to verify our results, we have run their approach again with and reduced the error tolerance parameter to 0.05 (the original default was 0.1) but it still does not perform satisfactorily. **We have provided new results for the subsampled Bank dataset in Figure 27 which we have added on Page 33 (Section R3) of our revised paper PDF**. It can be seen that in this instance with the smaller subsampled Bank dataset (training set has 100 samples, and test set is of the original Bank), influence-based data selection approaches perform better than TMC-Shapley. The code for these experiments is located here: <https://anonymous.4open.science/r/ICLR_Rebuttal_Experiments>.
> >
> >     ___
> >
> > * _**Instead of TMC-Shapely and random, it would have been more informative to see how the proposed approach compares with other influence estimation-based approaches, including vanilla (without CART) influence estimation.**_
> >
> >     Thank you for the concrete suggestion for improvement. To undertake this experiment, we have considered the original/vanilla influence computation approach by Koh and Liang, alongside our influence tree estimation implementation. We have also considered a more recent yet simplistic approach based on model self-confidence [2] that has been used to detect issues with training samples (such as label errors) as a baseline for comparison along with TMC-Shapley. **These new results are provided for the subsampled Bank dataset in Figure 27 which we have added on Page 33 (Section R3) of our revised paper PDF**. It is evident that our performance (red line) is superlative to the other baselines. The code for these experiments is located here: <https://anonymous.4open.science/r/ICLR_Rebuttal_Experiments>.
> >
> >     ___
> >
> >
> > * _**While the focus on convex loss is understandable, it might lead to sub-optimal influence value estimation due to the model parameters not being at a stationary point or the model not converging. This might then be a net negative and misleading data value estimation.**_
> >
> >     We agree with the reviewer, the fundamental limitation of the convexity assumption is that for non-convex models it is not possible to derive theoretical guarantees for the efficacy of influence functions. However, despite this lack of provable theoretical guarantee, by adding a damping term to the model's Hessian or using convex surrogates, our experiments on non-convex models such as MLPs and (preliminary results on) BERT in the appendix, we obtain significant performance improvements using our proposed approaches for multiple functions of interest. Despite this, as the reviewer pointed out there might be cases where they do not work as expected.
> >
> >     ___

---

> ### Author Response · Authors · 2023-11-17
> **Response to Reviewer MEKL [3/3]**
>
> * _**It looks like the authors do one utility at a time. Due to often competing utilities, for example, key features and samples for fairness might not necessarily be the same for accuracy, and in most cases might have a negative influence. It would be interesting to see an interplay of various utilities.**_
>
>     Thank you for the great suggestion, we can do this using our current obtained results. By observing the joint overlapping influence regions (or influence tree rules) for multiple functions of interest we can assess what samples contribute positively or negatively to both the functions of interest being considered (fairness, utility, etc.). In the current version of our paper, we did not do this since visualization for this is somewhat non-trivial and messy; however, in practice this is easy to do using our approach. **To exemplify this, we provide new analysis for the synthetic toy dataset of Section 4, for both accuracy + fairness in Figure 25 (Section R1, Page 32) of our revised paper PDF, and for accuracy + robustness in Figure 26 (Section R2, Page 32) of our revised paper PDF**. It can be seen that there are certain samples that are beneficial to remove for both accuracy and robustness, but for fairness and accuracy there is no overlap. Through this, we would like to emphasize that making joint improvements along multiple functions of interest is a dataset-dependent property and at times it might not be possible to make improvements along both functions (for e.g. the fairness-accuracy tradeoff [3]).
>
>     ___
>
> * _**Although authors use several datasets, all of them are binary settings. Value computation increases with classes, so I am curious to know if this is the reason authors only focused on binary settings or if there is another reason behind this design choice.**_
>
>     The choice for utilizing binary classification datasets are based on three factors. **First**, a number of approaches and work in fairness explicitly consider binary classification [3-5] as it is easier to interpret outcomes, and since we wanted to compare with these fairness baselines, we sought to utilize binary datasets throughout the paper. For example for works in both Section 5.1 (fairness distribution shift) and Section 5.2 (fairness poisoning attacks) datasets have binary classes. **Second**, we believe and hope that since our approaches are general, the binary case can be empirically extended to the multi-class case in future work without too much of a challenge. **Third**, a minor consideration was to have _neater_ looking trees when visualized. In the binary case, the class label can only take on two values, but in the multi-class case it would take on many different combinations and interplay with features in numerous ways leading to more complex trees.
>
>
> ___
> ___
>
> **References:**
> 1. Su, Jiang, and Harry Zhang. "A fast decision tree learning algorithm." AAAI 2006.
> 2. Northcutt et al. "Confident learning: Estimating uncertainty in dataset labels." Journal of AI Research 2021.
> 3. Li, Peizhao, and Hongfu Liu. "Achieving fairness at no utility cost via data reweighing with influence." ICML 2022.
> 4. Roh et al. "Improving fair training under
> correlation shifts". ICML 2023.
> 5. Roh et al. "Fairbatch: Batch selection for
> model fairness." ICLR 2021.

---

> ### Comment · Reviewer_MEKL · 2023-11-19
> **General reponse to authors.**
>
> I would like to express my gratitude to the authors for addressing the concerns raised by me and other reviewers.
>
> After reviewing the rebuttal responses and the anonymous repository, I conducted experiments to verify specific responses on runtime, TMC-Shapely and random deletion results, and data split questions.
>
> Upon analyzing the Bank data from the authors' anonymized repository (https://anonymous.4open.science/r/ICLR_Rebuttal_Experiments/), I discovered that the data separation did not align with the expected train:80/val:20 split. The train data had a shape of (100, 51), while the test data had a shape of (6098, 51).
>
> I ran several experiments with an error rate of 0.1, including a single run with seed 0 and merged runs of seeds [0, 1, and 2]. The results showed that removing the high-value TMC-Shapely data points decreased the accuracy (graphs labeled hightolow) while removing the low-value TMC-Shapely data points increased performance (graphs labeled lowtohigh). The results were consistent across both cases.
>
> When I used random deletion, the utility values were not linear. Lastly, the single run took 288.0850269794464 seconds, while the three runs altogether took 1477.8369567394257 seconds.
> In the single run (data_val_rebuttal_v2.ipynb/rebuttal_results2), I used computed TMC-Sahpley, random, and LOO data values, which means the time would have been even smaller. For the 3 runs (data_val_rebuttal.ipynb/rebuttal_results), I computed TMC-Shapley, G-Shapley, random, and LOO data values.
>
> The results and all the code are in the anonymized repository (https://anonymous.4open.science/r/iclr-85E3).
> The repository contains the data valuation scripts, experimentation notebooks, PKL files, and PDFs.
>
> Lastly, it looks like from this  ``[0.025, 0.05, 0.075, 0.1, 0.125]`` authors delete ``[int(i*len(x_train)) for i in l] = [2, 5, 7, 10, 12]`` data samples (not percentages as indicated in the figure). In my opinion, the percentage of data deleted in the experiments (up to 5% in other figures) is very small to properly assess/compare the algorithms. Additionally, in cases like the random method where the deletion of high-value data points might show the expected trend (worse utility), the deletion of low-value data points might not do as well, as expected. So it might be intuitive to see data deletion (both ways) to properly assess the effectiveness of the algorithm.
>
> I am happy to engage with the authors on these observations and or if I made some mistakes (it's possible).

---

> > ### Author Response · Authors · 2023-11-20
> > **Additional Response to Reviewer MEKL [1/2]**
> >
> > Dear Reviewer MEKL, thank you for your prompt response and evaluating the newly provided code. We appreciate your efforts in helping strengthen our work.
> >
> > We believe there might be some misunderstanding about our dataset/experiments (on time complexity for instance-- Table 2 in the main paper) which we will aim to clarify below. We apologize for any inconvenience due to a lack of clarity on our part. We also would like to thank the reviewer for their excellent code which we have now used to provide clarifications.
> >
> > * **Time Complexity Experiments**:
> >     * Our time complexity results in the paper (Table 2 in Appendix E) and the results comparing training sample deletions for our influence approach and TMC-Shapley (Figure 10 in Appendix E) are NOT undertaken on the same (Bank) dataset.
> >     * For all the results in the entire paper (Figures 2, 5, 6, 9, and others) we have considered full versions of the Bank, CelebA, Adult, and Jigsaw Toxicity datasets.
> >     * However, as Table 2 shows, TMC-Shapley was unable to run in reasonable time on these full datasets, and hence, we resorted to doing a simple experiment on a subsampled training set for Bank (only 100 samples), but the test set was kept unchanged. We also mentioned this in Appendix E.2, but we will aim to make this point much more clearly.
> >     * This subsampled version of the Bank dataset is the one we provided to the reviewer since TMC-Shapley was unable to run on the full versions that we did all our experiments on. This is also why the reviewer noted that the 80-20% split was not adhered to, and that the shape of the training set was (100, 51) and test set was (6098, 51).
> >     * We have now used the reviewer's excellent code and provided our original data files for all datasets here: <https://anonymous.4open.science/r/time_complexity_iclr-EB0A/>. In this repository, we provide the experiments on time complexity as a notebook (please refer to `time_complexity.ipynb`), and it can be seen that TMC-Shapley fails to run in a reasonable time on these full versions of the datasets.
> >     * We would also like to mention that our first original code repository provided in the paper: <https://anonymous.4open.science/r/IDS/> contains all the experiments for our approach where we experimented on the full datasets. The sizes of these datasets are as follows: Bank (30488), Adult (45222), CelebA (104163), and Jigsaw Toxicity (30000). The split details are also provided in Appendix C.1.
> >
> > ___
> >
> > * **Random Deletion Experimental Results**:
> >     * We would like to mention that in the reviewer's provided code, random deletion is undertaken till up to 40% of the dataset. In contrast, our experiments in the main paper (Figure 2 and others such as Figure 8 and 9) all focus on 5% data "trimming." Here our goal was to show how our influence based trimming approach can be extremely advantageous even for small % of data removals especially on full sized datasets.
> >     * In the specific experiments of Appendix E.2 we decided to trim up to 12.5%. Even for the reviewer's obtained results for random deletion that match with this setting on the subsampled Bank dataset (`rebuttal_results2/dval_10_lowtohigh`) it can be seen that random deletion is almost a constant line up till 10-12%.
> >     * The reason for a low removal % in our main experiments on the full datasets is that even 5% of removal often leads to desired performance: for example in Figure 2, Bank and CelebA reach DP (fairness) values close to 0 around 1% or 2% and hence, further trimming is not necessary. Similarly for a lot of the accuracy/robustness experiments, no further improvements in utility can be obtained after a certain point, indicating that we have reached the desired upper bound for accuracy. For fair comparison with our approach, we also randomly delete samples only up to 5%.
> >
> > ___

---

> > > ### Author Response · Authors · 2023-11-20
> > > **Additional Response to Reviewer MEKL [2/2]**
> > >
> > > * **TMC-Shapley Experimental Results**:
> > >     * We would like to thank the reviewer for their provided code. It seems that for some different seed values, indeed TMC-Shapley can work as intended (somewhat similar trends as the reviewer obtained), which we unfortunately did not see originally for our single seed experiment.
> > >     * We realized that another reason for the unsatisfactory result was possibly a mismatch in the logistic regression classifier; we use the one for our influence computation (we used PyTorch to train the model) to evaluate accuracy upon deletion/retraining, but this is different from the one used by TMC-Shapley to do data valuation in the backend. As a result, the mismatch might have been leading to slight errors. We  thank the reviewer for their efforts in providing code which led us to think about why this might be happening more critically.
> > >     * Thus, we will mention and showcase in Section E.2 of the paper (after working to minimize the classifier mismatch) that TMC-Shapley can work well but there might be inconsistencies in comparison (such as in the classifier implementation used and seed values considered). We will provide trends averaged over multiple seeds for TMC-Shapley for this reason. However, our main point stands that TMC-Shapley does not run in reasonable time on the full sized datasets we consider in the paper and we would like to emphasize this as well.
> > >
> > > ___
> > >
> > > * **Percentages in Deletion and Two-way Deletion**:
> > >     * We are deleting percentages in an automated fashion, but as is evident if the percentage leads to a fractional value, we make them integral as the number of points to be deleted has to be an integer value.
> > >     * For our main experiments (Section 4) and non-conventional classification experiments (Section 5), even 5% data deletion can be a large number of samples, in contrast with the subsampled Bank dataset experiment of Appendix E.2 which was just a simple experiment with 100 training samples. For example, for CelebA, 5% deletion implies deleting 3124 samples (5% of 62497 training samples).
> > >     * Regarding two-way deletion (high influence and low influence samples), we agree with the reviewer's great suggestion. Given that we have a large number of experiments in the paper and the dataset sizes are quite large, re-running all experiments would be quite time consuming. If the reviewer would like to see a specific experiment with high influence sample removal, we are happy to explore it and try to finish it within the author-reviewer discussion period. Also thematically, our applications considered in this paper are geared towards improving performance and hence, deleting low influence samples makes more sense. Our primary focus in this work is also not just influence trimming, but only using it as a tool for one facet of the paper and showcasing how feature ranges can be interpreted for improved performance in diverse settings (such as active learning and online learning). We will definitely aim to study high influence sample deletion in future work.
> > >
> > > ___
> > > ___
> > >
> > > Thank you once again for all your efforts, we are very grateful.

---

> > > > ### Comment · Reviewer_MEKL · 2023-11-20
> > > >
> > > > I appreciate the authors for their detailed responses, addressing concerns, and correcting misunderstandings. I have decided to increase my score.

---

> > > > > ### Author Response · Authors · 2023-11-20
> > > > >
> > > > > Dear Reviewer MEKL, thank you for your valuable efforts and time in discussing with us, and aiding us in improving our codebase and contributions. We appreciate it.
> > > > >
> > > > > Kind Regards,
> > > > >
> > > > > Authors.

---

### Official Review · Reviewer_MiDf · 2023-11-01

**Soundness:** 3 good
**Presentation:** 3 good
**Contribution:** 3 good
**Rating:** 6
**Confidence:** 4

**Summary:**

This paper utilizes influence functions to assess what data samples improve utility (smaller loss), fairness (DP and EOP), and adversarial robustness for a given convex classifier by interpreting which sample features contribute positively or negatively to certain performance metrics, and design a data selection strategy accordingly.

**Strengths:**

1.	Consider many aspects of model performance beyond accuracy; especially the fairness.
2.	Experiments are thorough and the presentations of the experimental results are sound and clear.

**Weaknesses:**

1.	Limitation on model class: The authors provide a discussion on why the influence function evaluations are limited to convex classifiers, possible remedies, and recently applications to deep neural networks. However,
2.	Theoretical analysis: the estimation of the influence function is based on the trees with hierarchical shrinkage regularization. However, there is no analysis on the credibility, time complexity of the proposed Algorithm 1 and Algorithm 2. It seems that these algorithms are not scalable to large-scale datasets.
3.	The utility, fairness and adversarial robustness are important performance metrics for a classifier; however, there is a lack of a unifying story to connect all three and therefore the discussion and experiments may seem distracted
4.	Feature explanation is a key aspect in this paper; however, the connection of feature explanation using the influence function with existing explainable AI literature is lacking.

**Questions:**

1.	Should not the influence estimator has the same architecture of the classifier?
2.	For the fairness experiments in Section 5.1, would the authors justify the choice of the fairness intervention baselines?
For other questions, please refer to the Weaknesses. I will consider raising the scores if the authors could adequately address my questions in the rebuttal.

---

> ### Author Response · Authors · 2023-11-17
> **Response to Reviewer MiDf [1/2]**
>
> We would like to thank the reviewer for their detailed review of our work, and are grateful for their time and effort. Below we provide answers to the weaknesses and questions raised:
>
> * **Response to Weaknesses**:
>
>     - > Limitation on model class: The authors provide a discussion on why the influence function evaluations are limited to convex classifiers, possible remedies, and recently applications to deep neural networks. However,
>
>         It seems that perhaps this question was accidentally cut-off. If the reviewer can provide more details, we will aim to provide a targeted response and justification during the discussion phase.
>
>     ___
>
>
>     - > Theoretical analysis: the estimation of the influence function is based on the trees with hierarchical shrinkage regularization. However, there is no analysis on the credibility, time complexity of the proposed Algorithm 1 and Algorithm 2. It seems that these algorithms are not scalable to large-scale datasets.
>
>         Both our proposed algorithms are very computationally efficient and can scale with large datasets. We provide more analytical details on time complexity below.
>
>         **Algorithm 1 Time Complexity**: For calculating computational complexity, we need to analyze the two steps of our influence estimation process. First, the influence computation of training samples and a given base model requires $\mathcal{O}(npm)$ time where $n,p$, and $m$ are number of training samples, model parameters, and test samples respectively. Please refer to Section 3 of Koh and Liang's paper for how influence can be computed in $\mathcal{O}(npm)$ using stochastic estimation or conjugate gradients. Second, the cost of training time of the influence estimation tree in the worst case is $\mathcal{O}(n^2 d)$ where $d$ is the number of features. Thus, the overall complexity is $\mathcal{O}(npm) + \mathcal{O}(n^2 d)$. Note that the quadratic dependence of the second term can be made linear in $n$ simply be using more sophisticated tree-building methods [1] which have a time complexity of $\mathcal{O}(nd)$. Since the overall time complexity will be linear, the approaches proposed are amenable to large scale datasets (CelebA in our experiments has 104K samples). Furthermore, our approaches work with very high-dimensional data as well. As shown in experiments on the image (CelebA) and NLP (Jigsaw Toxicity) datasets we consider in the paper, we utilize embeddings obtained by deep learning models as a preprocessing step prior to our approach. This significantly reduces the dataset feature size, and at the same time allows for a rich and powerful feature space that can be used to train our models/methods.
>
>         **Algorithm 2 Time Complexity**: The time complexity of Algorithm 2 is easier to calculate. Note that the influence values provided once at input require $\mathcal{O}(npm)$ to calculate as mentioned before. Then the sorting step requires $\mathcal{O}(n log(n))$ in the worst case. The other operations (lines 8-9) are linear vector operations, leading to a worst case time complexity of $\mathcal{O}(n(pm + log(n))$. This is not a computational bottleneck for large datasets as the time complexity is linear or logarithmic for each factor.
>
>     ___
>
>     - > The utility, fairness and adversarial robustness are important performance metrics for a classifier; however, there is a lack of a unifying story to connect all three and therefore the discussion and experiments may seem distracted
>
>         Thank you for the great suggestion, we can do this using our current obtained results. By observing the joint overlapping influence regions (or influence tree rules) for multiple functions of interest we can assess what samples contribute positively or negatively to both the functions of interest being considered (fairness, utility, etc.). In the current version of our paper, we did not do this since visualization for this is somewhat non-trivial and messy; however, in practice this is easy to do using our approach. **To exemplify this, we provide new analysis for the synthetic toy dataset of Section 4, for both accuracy + fairness in Figure 25 (Section R1, Page 32) of our revised paper PDF, and for accuracy + robustness in Figure 26 (Section R2, Page 32) of our revised paper PDF**. It can be seen that there are certain samples that are beneficial to remove for both accuracy and robustness, but for fairness and accuracy there is no overlap. Through this, we would like to emphasize that making joint improvements along multiple functions of interest is a dataset-dependent property and at times it might not be possible to make improvements along both functions (for e.g. the fairness-accuracy tradeoff [2]).
>
>
>     ___

---

> > ### Author Response · Authors · 2023-11-17
> > **Response to Reviewer MiDf [2/2]**
> >
> > * **Response to Weaknesses**: (Continued)
> >
> >     - > Feature explanation is a key aspect in this paper; however, the connection of feature explanation using the influence function with existing explainable AI literature is lacking.
> >
> >         Thank you for the great suggestion for strengthening the paper. We will add a paragraph for discussion on this in the main paper as follows:
> >
> >         _Most explainable and interpretable ML/AI approaches leverage feature importances to understand why models decided on a particular class label for a sample. For instance in [3], the goal is to obtain what regions of the image contribute most to the model picking that label for the given image. This interpretability by means of explaining sample-level model predictions contrasts with our work on interpretability, where we aim to interpret what samples/features benefit the model or lead to its detriment with respect to a function of interest (fairness, utility, etc.)._
> >
> >     ___
> >     ___
> >
> > * **Response to Questions**:
> >
> >     - > Should not the influence estimator has the same architecture of the classifier?
> >
> >         Since we would like to obtain interpretable rule sets/decisions for what features contribute positively or negatively to the function of interest, we opt for a decision tree regressor. However, since the learning problem is in general a regression problem where the inputs are sample feature, label tuples and targets are influence values, any interpretable classifier can be used. It is not necessary to utilize the same model or architecture as the original base model.
> >
> >     ___
> >
> >     - > For the fairness experiments in Section 5.1, would the authors justify the choice of the fairness intervention baselines? For other questions, please refer to the Weaknesses. I will consider raising the scores if the authors could adequately address my questions in the rebuttal.
> >
> >         The baseline approaches for comparison in Section 5.1 were selected based on multiple factors. **First**, due to the vast literature on fairness, we aimed to select methods that were recently proposed (2020-2023). **Second**, our aim was to compare with approaches that belonged to the preprocessing category of fairness interventions, since our approach is also preprocessing based. Except for the Robust Fair Logloss Classifier (RLL) which is an in-processing approach, all baselines belong to this category. **Third**, our aim was to select approaches that have been specifically used for mitigating fairness distribution shift or have robustness properties so as to counter fairness shift. All the approaches considered (Correlation Shift, Influence-based Reweighing, FairBatch, and RLL) meet this criteria. Additionally, Influence-based Reweighing is a recent approach from 2022 that also utilizes influence functions, so our aim was to show how our proposed approach can improve performance even more when used in conjunction with it. We are happy to add more as well if the reviewer would like to point out any key missing approaches for comparison.
> >
> > ___
> > ___
> >
> > **References:**
> > 1. Su, Jiang, and Harry Zhang. "A fast decision tree learning algorithm." AAAI 2006.
> > 2. Li, Peizhao, and Hongfu Liu. "Achieving fairness at no utility cost via data reweighing with influence." ICML 2022.
> > 3. Chen, Chaofan, et al. "This looks like that: Deep learning for interpretable image recognition." NeurIPS 2019.

---

> > > ### Comment · Reviewer_MiDf · 2023-11-22
> > >
> > > I appreciate the authors for their response. For my question that was accidentally cut off, I found the answer to it in the response to other reviewers. As the authors have addressed my concerns, I decided to raise my score.

---

> > > > ### Author Response · Authors · 2023-11-23
> > > > **Additional Response to Reviewer MiDf**
> > > >
> > > > Dear Reviewer MiDf,
> > > >
> > > > Thank you for your response and checking other reviewers' comments. We are happy to know you can find that answer in the response to other reviewers, which well addressed your major concern.
> > > >
> > > > Again thank you for your  efforts and time in reviewing our paper and valuable comments. Appreciated!
> > > >
> > > > Kind Regards,
> > > >
> > > > Authors.

---

### Official Review · Reviewer_GBD9 · 2023-11-05

**Soundness:** 3 good
**Presentation:** 3 good
**Contribution:** 3 good
**Rating:** 6
**Confidence:** 4

**Summary:**

This paper presents a new approach to enhancing the performance of classification models by interpreting and selecting training data through influence estimation models. The authors aim to improve model utility, fairness, and robustness by identifying data that positively impacts these aspects. Extensive experiments on various datasets demonstrate the effectiveness of their methods, which are also applicable to scenarios like distribution shifts and fairness attacks.

**Strengths:**

1. **Important Research Problem**: The authors have targeted an important research problem that focuses on selecting important training data to improve model performance. The research can improve the effectiveness of machine learning models' development that is often overlooked in favor of more complex model architectures or algorithms.

2. **Thorough Experiments**: The authors have conducted thorough experiments to validate their approaches. The use of both synthetic and real-world datasets ensures that the findings are robust and not limited to specific types of data or scenarios. This comprehensive testing framework strengthens the validity of the research conclusions.

3. **Many Applications**: One of the paper's strengths lies in its application to different scenarios. The authors have not only considered conventional classification tasks but have also extended their methodology to address other challenges such as distribution shifts, fairness poisoning attacks, utility evasion attacks, online learning, and active learning. This broad applicability demonstrates the potential impact of the research on various domains and highlights the versatility of the proposed methods.

**Weaknesses:**

1. **Scalability Concerns**: The use of tree-based influence estimation models might indeed pose scalability issues. Tree-based models can become computationally expensive as the size of the dataset increases, especially if the influence estimation requires building trees for many subsets of data or for complex feature interactions. This could limit the method's applicability to big data scenarios or require significant computational resources, which may not always be feasible.

2. **Hard to Adopt Data with High-Dimensional Features**: For example, image data presents unique challenges due to its high dimensionality and the spatial relationships between pixels. Influence functions and feature space interpretations that work well for tabular data may not translate directly to image data.

**Questions:**

- How do the tree-based influence estimation models proposed by the authors scale with very large datasets, and what are the computational costs associated with these models?
- Could the authors provide insights into the computational complexity of their influence estimation approach, and are there any strategies they recommend for scaling it to big data applications?
- How does the tree model handle high-dimensional data, such as images, where feature interactions are more complex?
- Could the authors elaborate on any modifications or extensions to their approach that might be necessary to apply it effectively to image data or other high-dimensional datasets?
- The work presented focuses on convex models or convex surrogates for non-convex models. Could the authors discuss the potential limitations of this?

---

> ### Author Response · Authors · 2023-11-17
> **Response to Reviewer GBD9**
>
> We would like to thank the reviewer for their thoughtful review and for all their time and effort spent in reviewing our work. Below we answer the questions raised:
>
> * _**How do the tree-based influence estimation models proposed by the authors scale with very large datasets, and what are the computational costs associated with these models?**_
>
>     The tree-based influence models are computationally efficient and can scale with large datasets. They have a training time of $\mathcal{O}(npm) + \mathcal{O}(n^2 d)$ in the worst case where $n$, $p$, $m$, and $d$ are the number of training samples, model parameters, test samples, and features, respectively. The first term is for the influence computation step (please refer to Section 3 of Koh and Liang's paper for how influence can be computed in $\mathcal{O}(npm)$ using stochastic estimation or conjugate gradients) and the second is for training and constructing the tree model. The biggest factor here is the quadratic dependence on the number of training samples for constructing the tree, which can be made linear in $n$ simply by using more sophisticated tree-building methods [1], which have a time complexity of $\mathcal{O}(nd)$. This will result in an overall time complexity of $\mathcal{O}(n(pm + d))$. Thus the model can scale easily to larger datasets on modern hardware. This is also evident in our experiments in the paper on the CelebA dataset that has 104K samples and runs computationally efficiently even with a standard tree-building algorithm such as CART (it takes 2.455 seconds to run on average on a Linux server with an Intel Xeon 2.2GHz CPU).
>
>     ___
>
> * _**Could the authors provide insights into the computational complexity of their influence estimation approach, and are there any strategies they recommend for scaling it to big data applications?**_
>
>     Our approaches can be used with big data. For calculating computational complexity, please refer to the previous response. As mentioned before, even though the overall time complexity is $\mathcal{O}(npm) + \mathcal{O}(n^2 d)$, using more sophisticated tree-building methods [1] we can improve time complexity to $\mathcal{O}(n(pm + d))$ which is linear in $n$. Furthermore, for very high-dimensional data ($d$ is large), a feature extraction step can reduce time complexity considerably. For the image (CelebA) and NLP (Jigsaw Toxicity) datasets we consider in the paper, we utilize embeddings obtained by deep learning models as a preprocessing step prior to our approach. This significantly reduces the dataset feature size, and at the same time allows for a rich and powerful feature space that can be used to train our models/methods.
>
>     ___
>
> * _**How does the tree model handle high-dimensional data, such as images, where feature interactions are more complex? Could the authors elaborate on any modifications or extensions to their approach that might be necessary to apply it effectively to image data or other high-dimensional datasets?**_
>
>     Our approaches can be used efficiently with high-dimensional image and text datasets simply by employing a feature extraction preprocessing step on the raw data such as by obtaining embeddings from a deep learning model or some other approach. This is how we conduct experiments on CelebA and Jigsaw Toxicity datasets. For e.g., we use the Mini-LM transformer model to get embeddings for the Jigsaw Toxicity dataset and then conduct experiments on it. This significantly reduces the dataset feature size which further reduces the overall worst case time complexity. Additionally, we would also like to mention that past work has found that pixel-level interpretation of images is not akin to human interpretation/reasoning, and hence, prototype-based explanations are generally preferred [2]. Hence, for raw image data, using a feature extraction step for preprocessing is a viable option for improved interpretation.
>
>     ___
>
> * _**The work presented focuses on convex models or convex surrogates for non-convex models. Could the authors discuss the potential limitations of this?**_
>
>     The most fundamental limitation of the convexity assumption is that for non-convex models it is not possible to derive theoretical guarantees for the efficacy of influence functions. However, despite this lack of provable theoretical guarantee, in practice this does not invalidate the use of influence functions in non-convex models. By adding a damping term to the model's Hessian or using convex surrogates, influence functions can obtain satisfactory results. For instance, in our experiments on non-convex models such as MLPs and (preliminary results on) BERT in the appendix, we obtain performance improvements using our proposed approaches for multiple functions of interest.
>
> ___
> ___
>
> **References:**
>
> 1. Su et al. "A fast decision tree learning algorithm." AAAI 2006.
> 2. Chen et al. "This looks like that: Deep learning for interpretable image recognition." NeurIPS 2019.

---

### Official Review · Reviewer_bDe3 · 2023-11-10

**Soundness:** 3 good
**Presentation:** 3 good
**Contribution:** 3 good
**Rating:** 6
**Confidence:** 4

**Summary:**

This paper presents two schemes for 1) identifying the effect of a training point on a function of a model's parameters, and 2) trimming the training set to stem the influence of training instances that have a high negative influence on the test metric of interest. The paper examines the influence of a training point on a test set fairness metric, adversarial robustness, and utility. The first algorithm fits a regression tree using the input and label to the influence function of a metric of interest. The second algorithm then trims the subset of the training set to improve the model. The paper then demonstrates this approach across a variety of metrics including mitigating the effect of unfairness due to distribution shift, adversarial robustness, and the effect of noisy labels in the streaming setting across several datasets.

**Strengths:**

Overall, this paper provides a comprehensive empirical demonstration of how to improve a performance metric of interest given training samples and model parameters.

- **The breadth of properties**: This paper considers several interesting scenarios ranging from noisy labels, active learning, adversarial robustness to fairness. The comprehensive nature of these settings is quite impressive and commendable.
- **Compares to adequate baselines**: The paper considers the key baselines that I would've expected and shows improved performance over these baselines.
- **Compelling results**: I particularly like Figure 2. Over a range of properties and settings, we see that influence-based deletion approach remains quite effective.

**Weaknesses:**

I have a number of confusion about this work that I state here and in the questions section. I would be happy to revise my score in light of feedback from the authors.

- **Details of the approach**: The exact procedure of the trimming portion is not quite clear to me. I think the authors miss discussing retraining. I assume that the authors are referring to a model retrained after a subset of examples are deleted? So in Fig. 2, x axis==zero is a model trained on all data points? Then you trim a percent of the training samples and then retrain a model on the new dataset? If yes, is it the original model that is used for deciding which samples to trim or is the model changing?

- **What is the motivation behind the cart regression procedure?**: As it stands it seems the cart procedure takes as input $(x_i, y_i)$, and the predicts some influence score per example? More details could be useful here. Are the samples used in the training of the cart model a subset of the original training set for which the influence was estimated? Since we know that the influence score measures the effect of up(down)weighting the training sample, alone, we also know that the label should not have any effect on predictive quality of the tree. What is the point of then concatenating the label? It seems like the goal here is to estimate the effect of a feature on the performance metric of interest. I take this judging from Figure 1 where the authors plot performance metric vs features that is colored by influence. If the goal is really to determine the effect of a feature on the performance metric of interest, then how to do that is already in section 2.2 of the original Koh and Liang paper. If the goal is not to measure the effect of the feature on the influence score, then I am not sure I understand the point of this section. Another point here is that in the rest of the paper, the trimming-based approach is really what the authors use and not the cart procedure. If this is the case, I don't think we can that as a contribution of this work. I am asking all these questions as a way to better understand the motivation and goal of fitting the tree to predict the estimated influence score.

- **Related Work**: There is some related work that this paper should be aware of. I list them here: Kong et. al., Resolving Training Biases via Influence-based Data Relabeling, Adebayo et. al. Quantifying and mitigating the impact of label errors on model disparity metrics, Richardson et. al. Add-Remove-or-Relabel: Practitioner-Friendly Bias Mitigation via Influential Fairness, (concurrent) Understanding Unfairness via Training Concept Influence, Sattigerri et. al. Fair infinitesimal jackknife: Mitigating the influence of biased training data points without refitting. All of these papers have a trimming and/or relabelling scheme in them. I am not claiming that this work is not novel/important. I think the insights here are quite useful actually, but it would be helpful for the authors to acknowledge these works, and contextualize their contributions in light of these papers.

- **Tabular Data**: I don't see this as an important weakness; however, most of this work is demonstrated on tabular data. It will be tricky to extend the feature analysis portion, as done in Figure 1 for example, to say images or text.

**Questions:**

Please see the first two bullet points in the weaknesses section for a list of the questions that I have. Thanks.

---

> ### Author Response · Authors · 2023-11-17
> **Response to Reviewer bDe3 [1/3]**
>
> We would like to thank the reviewer for their deep analysis of our work, and their time, effort, and consideration. Below, we answer the questions raised by the reviewer:
>
> * **Details of the approach**: We apologize for any lack of clarity in presenting the approach.
>
>     - > I assume that the authors are referring to a model retrained after a subset of examples are deleted?
>
>         Yes, the reviewer is correct.
>
>     - > So in Fig. 2, x axis==zero is a model trained on all data points?
>
>         Yes, in Figure 2, x-axis = 0 the model is trained on all data samples.
>
>     - > Then you trim a percent of the training samples and then retrain a model on the new dataset?
>
>         Yes, the reviewer is correct, this is precisely what we do.
>
>     - > If yes, is it the original model that is used for deciding which samples to trim or is the model changing?
>
>         The original unchanged model is used to decide which samples to trim and then we sequentially trim the samples to the desired percentage.
>
>
>     Thank you for these details. We will make the above points clearer in the text.
>
>     ___
>
> * **What is the motivation behind the CART regression procedure?**: Thank you for these insightful questions. We would first like to illustrate our motivation/goal and then address these questions one by one. Our goal is to discover a subset of the training samples that benefit the learning model (via influence functions to trim the samples that are a detriment to the model) and interpret where the benefits come from in the feature space (via CART regression).
>
>     - > Are the samples used in the training of the CART model a subset of the original training set for which the influence was estimated?
>
>         Yes, the reviewer is correct, we use an 80-20 split with the original samples (concatenated with their labels) for training the CART model that will then be used to predict the influence value.
>
>     - > Since we know that the influence score measures the effect of up(down)weighting the training sample, alone, we also know that the label should not have any effect on predictive quality of the tree. What is the point of then concatenating the label?
>
>         The label implicitly contributes to the computation of the influence function. Since the influence function estimation utilizes the gradient of the loss term $l$, which takes in as input both $x_i$ (training sample) and $y_i$ (class label), the label also contributes to the influence computation. This is evident in Eqs. (1-3) in our paper and Eq. (2) in Koh and Liang's paper, where they use $z_i$ which is a tuple = $(x_i, y_i)$. This is why we concatenate the label for influence estimation.

---

> > ### Author Response · Authors · 2023-11-17
> > **Response to Reviewer bDe3 [2/3]**
> >
> > * **What is the motivation behind the CART regression procedure?**: (Continued)
> >
> >     - > If the goal is really to determine the effect of a feature on the performance metric of interest, then how to do that is already in Section 2.2 of the original Koh and Liang paper.
> >
> >         - The difference is that in Koh and Liang's paper (Section 2.2) they have provided an approach for computing influence at the sample-level whereas we are trying to derive general "rule sets" at the feature-level for how features affect influence. For example, in Koh and Liang's work, they can obtain influence $I(x_i)$ for a sample $x_i$ (or change in output with sample-level perturbations). In our work, we use a large set of such samples (and their labels as mentioned previously) and their influence values, to train an interpretable estimator, such as CART. This can then be used to derive a general interpretation for influence using the feature space-- for example some rules for Accuracy from our 2-D feature space in Figure 1 are: Feature 1 $>= 1.9$ and Feature 2 $< 0.1$ lead to high influence while Feature 1 $< 1.9$ and Feature 2 $< 0.1$ lead to low influence.
> >
> >
> >         - This sort of feature-level interpretation of performance gain via influence has not been undertaken in the literature previously. This can be useful in many scenarios, some of which we outline below (but are not limited to only these):
> >
> >             **(1)** When new data needs to be added that we do not have influence scores for yet, we can use the influence estimator to obtain influence scores simply via inference. For Koh and Liang's approach, they would still need to train the model once with all samples under consideration every time a new sample arrives.
> >
> >             **(2)** In algorithmic data recourse scenarios [1] such as to describe why an individual was denied a loan, it can be used to provide an interpretable justification for why a decision was made simply by describing the rules that were either met or violated by the data sample.
> >
> >             **(3)** In active learning, traditional influence computation (such as using Koh and Liang's work) is not viable since there is no access to the sample labels. Previous work such as ISAL [2] (one competitive method in our paper) have still utilized influence in these scenarios, but these do not work as well, since they use the base model itself to generate pseudo-labels for prediction. Our approach using CART serves as an effective and much better alternative, as is evident in the active learning experiments of our paper (Section 5.5 - Figure 6E), where we compare with ISAL. Note here we have to train the CART estimator without labels.
> >
> >             **(4)** In some sense, we are bridging feature-level interpretation approaches (such as Shapley values and LIME) with sample-level interpretation approaches (such as Data Valuation and Influence), and hence, can accommodate any applications that these are used for.
> >
> >
> >     - > Another point here is that in the rest of the paper, the trimming-based approach is really what the authors use and not the CART procedure.
> >
> >         - The trimming-based approach can be used for performance gain, but for the interpretation of performance gain, the CART procedure is needed (See Figure 3 and Appendix F). Moreover, we use the CART approach for our active learning experiments of Section 5.5 to estimate the influence of samples in the unlabeled pool set. Observing these results (also the additional ones in the appendix for the same) it is evident that our approach outperforms multiple other baselines, and is beneficial. Moreover, we believe Algorithm 1 provides qualitative practical value: for example, as we do in Section 5.5, a practitioner could use the decision tree model to interpret what samples are actually beneficial for active learning, and hence provide justification (to annotators and other stakeholders) for why it was picked for labeling. As mentioned before, another example is for a loan approval model where Algorithm 1 can be used to easily derive reasons (citing training set feature values) for why an applicant was rejected. This might even shed light on incorrect decisions being made by the model.
> >         - Examples of such influence estimation trees (i.e. interpretable rule sets) derived from Algorithm 1 can be seen in Appendix F. There is a huge body of literature on interpreting model's predictions, but  to our best knowledge, interpreting model's performance gain at the feature-level using influence has not been undertaken yet.
> >
> >
> > ___

---

> > > ### Author Response · Authors · 2023-11-17
> > > **Response to Reviewer bDe3 [3/3]**
> > >
> > > * **Related Work**: Thank you for bringing these papers to our notice, it is highly appreciated and will strengthen the discussion in our paper. We will cite these and include discussion on them in the main text. We contextualize them with regards to our approaches below:
> > >
> > >     * A major difference between our work and [3-7] is that none of these works focus on feature-level interpretability approaches that can allow practitioners to understand and interpret what feature ranges benefit the model or lead to its detriment. There are other differences as well, which we discuss individually below.
> > >
> > >     * [3,4,5]: In [3], authors focus specifically on relabeling schemes for harmful training samples using influence functions. While thematically related to improving model performance using influence, the approach taken in our work contrasts with theirs in terms of overall motivation and generality of methods. Our data selection methods and feature-level interpretations via influence are extremely general and simple, and can hence be applied to fairness, accuracy, robustness, and many scenarios such as distribution shift, adversarial attacks, active learning, etc. The authors in [4,5] extend the ideas proposed in [3] to fairness metrics, with [5] also allowing for addition and removal of samples specific to the profile of the practitioner. However, it is important to note that based on [3] and [4], different labeling functions need to be proposed for different functions of interest (fairness, utility, etc.) whereas with our approach this issue does not arise due to its simplicity and generality.
> > >     * [6]: This work specifically studies fairness of samples by generating counterfactual samples and then measuring their influence. While this work can aid understanding of the fairness of training samples, its motivation is different from our work, which seeks to extend and apply influence functions for model performance gains across multiple metrics of interest and for feature-level interpretability.
> > >     * [7]: In this work authors propose a downweighting scheme for training samples to study improvements in fairness for a number of fairness metrics. In spirit, this work is close to ours on influence-based data selection. However, the approaches proposed are more complex and specific to the study of fairness, whereas we opt for simpler and general approaches that can be widely applied in many settings, even beyond traditional classification.
> > >
> > >     ___
> > >
> > > * **Tabular Data**: We understand the reviewer's concern. We would like to point out that while 3 of the 5 datasets we consider are natively tabular datasets, the remaining 2 are vision/image (CelebA) and text/NLP datasets (Jigsaw Toxicity). Our approaches can be used with image and text datasets by using a feature extraction step. That is, feature extraction from the raw data needs to be undertaken as a preprocessing step using embeddings from a deep learning model or some other approach. This is how we conduct experiments on CelebA and Jigsaw Toxicity datasets. For example, we use the Mini-LM transformer model to get embeddings for the Jigsaw Toxicity dataset and then conduct experiments on it as if it were a tabular dataset. We would also like to mention that past work has found that pixel-level interpretation of images is not akin to human interpretation/reasoning, and hence, prototype-based explanations are generally preferred [8]. Hence, for raw image data, using a feature extraction step for preprocessing is a viable option for improved interpretation.
> > >
> > > ___
> > > ___
> > >
> > > **References:**
> > > 1. Ustun, Berk, Alexander Spangher, and Yang Liu. "Actionable recourse in linear classification." ACM FAT* 2019.
> > > 2. Liu, Zhuoming, et al. "Influence selection for active learning." CVPR 2021.
> > > 3. Kong, Shuming, Yanyan Shen, and Linpeng Huang. "Resolving training biases via influence-based data relabeling." ICLR 2021.
> > > 4. Adebayo, Julius, et al. "Quantifying and mitigating the impact of label errors on model disparity metrics." ICLR 2023.
> > > 5. Richardson, Brianna et al. "Add-Remove-or-Relabel: Practitioner-Friendly Bias Mitigation via Influential Fairness." ACM FaccT 2023.
> > > 6. Yao, Yuanshun, and Yang Liu. "Understanding Unfairness via Training Concept Influence." Arxiv 2023.
> > > 7. Sattigeri, Prasanna, et al. "Fair infinitesimal jackknife: Mitigating the influence of biased training data points without refitting." NeurIPS 2022.
> > > 8. Chen, Chaofan, et al. "This looks like that: Deep learning for interpretable image recognition." NeurIPS 2019.

---

> > > > ### Comment · Reviewer_bDe3 · 2023-11-20
> > > > **Addresses my concerns**
> > > >
> > > > Thanks to the authors for addressing my concerns. I now see what the point of the CART Regression procedure is. I agree with the authors about the point of doing the global feature analysis and that no previous work has done this. I just don't know that cart regression is the best way to do this kind of analysis. This is because such a procedure will be useful for tabular data, but for images and text, it clearly wouldn't be appropriate. The effectivenesses of your model will depend on how easy it is to interpret the features that is used to train the cart procedure. In addition, you also have to hope that the CART model is high performing and can learn useful features. This is my only remaining issue as it stands, but it is not a disqualifying one, and I think this work is compelling and has addressed my points.

---

> > > > > ### Author Response · Authors · 2023-11-21
> > > > >
> > > > > Dear Reviewer bDe3, thank you for the detailed review of our work and for your help in improving it further, we really appreciate it. We understand and agree with the reviewer on the aforementioned points regarding tabular data, although there is work that has been proposed recently (Hu et al, "Towards Deep Interpretable Features", 2023) that extracts interpretable features for downstream use (classification) with linear or convex models. Such methods could be used to extract interpretable features from raw image data, and then our approaches could be applied, further extending their use more appropriately beyond tabular datasets.
> > > > >
> > > > > Thank you once again for all your efforts in improving our submission, we are very grateful.

---

### Meta-Review · Area_Chair_grut · 2023-12-08

**Metareview:**

The paper presents an innovative data-selection approach that enhances model performance.

Key strengths highlighted by reviewers include: 1) addressing an important research problem; 2) considering various settings ranging from noisy labels, active learning, and adversarial robustness to fairness; 3) conducting thorough experiments with promising results; and 4) having numerous applications.

During the rebuttal, the authors made significant clarifications, and all reviewers acknowledged that all major concerns have been well addressed.

Given the positive feedback from reviewers, the AC recommends accepting this paper.

**Justification For Why Not Higher Score:**

NA

**Justification For Why Not Lower Score:**

Many strengths have been highlighted by reviewers, including addressing an important research problem, considering various settings, conducting thorough experiments, and having many applications. All reviewers acknowledged that all major concerns have been well addressed and provided positive ratings.

---

### Decision · Program_Chairs · 2024-01-16

Accept (oral)